# Joint, multifaceted genomic analysis enables diagnosis of diverse, ultra-rare monogenic presentations

Shilpa Nadimpalli Kobren [1,45], Mikhail A. Moldovan[1,45], Rebecca Reimers[2,3], Daniel Traviglia[1], Xinyun Li [4,5], Danielle Barnum[6,7], Alexander Veit[1], Rosario I. Corona[8], George de V. Carvalho Neto [8], Julian Willett[9], Michele Berselli [1], William Ronchetti[1], Stanley F. Nelson[8], Julian A. Martinez-Agosto[8], Richard Sherwood [7], Joel Krier[10], Isaac S. Kohane[1], Undiagnosed Diseases Network* & Shamil R. Sunyaev [1] ✉

Genomics for rare disease diagnosis has advanced at a rapid pace due to our ability to perform in-depth analyses on individual patients with ultra-rare diseases. The increasing sizes of ultra-rare disease cohorts internationally newly enables cohort-wide analyses for new discoveries, but well-calibrated statistical genetics approaches for jointly analyzing these patients are still under development. The Undiagnosed Diseases Network (UDN) brings multiple clinical, research and experimental centers under the same umbrella across the United States to facilitate and scale case-based diagnostic analyses. Here, we present the first joint analysis of whole genome sequencing data of UDN patients across the network. We introduce new, well-calibrated statistical methods for prioritizing disease genes with de novo recurrence and compound heterozygosity. We also detect pathways enriched with candidate and known diagnostic genes. Our computational analysis, coupled with a systematic clinical review, recapitulated known diagnoses and revealed new disease associations. We further release a software package, RaMeDiES, enabling automated cross-analysis of deidentified sequenced cohorts for new diagnostic and research discoveries. Gene-level findings and variant-level information across the cohort are available in a public-facing browser (https://dbmi-bgm.github.io/udn-browser/). These results show that case-level diagnostic efforts should be supplemented by a joint genomic analysis across cohorts.

For decades preceding the widespread application of DNA sequencing, identifying the genetic etiology of rare monogenic phenotypes including human diseases relied on segregation in pedigrees[1]. DNA sequencing enabled the analysis of sporadic cases with no segregation data[2]. Early studies analyzed small cohorts of phenotypically similar cases[3,4], a highly successful approach that is, however, limited to diseases with multiple known patients with fairly homogeneous presentations. In the absence of such phenotypically matched case cohorts, per-case studies of individual, undiagnosed patients are gaining popularity[5-8]. By design, these studies cannot attain statistical power from the shared genotypes of unrelated patients and require extensive clinical and biological inquiry to prove the causal

A full list of affiliations appears at the end of the paper. *A list of authors and their affiliations appears at the end of the paper.
✉e-mail: shamil_sunyaev@hms.harvard.edu

involvement of the genotype in disease[9–11]. The most recent phase of human Mendelian genetics employs a data science approach to gene discovery propelled by the joint genomic analysis of phenotypically broad cohorts. Recent studies by the Deciphering Developmental Disorders and 100,000 Genomes consortia have demonstrated the power of this approach to identify new diagnoses and disease genes[12–14]. This opens the prospect of international cross-cohort analyses, leveraging parallel efforts in many countries, and appreciating that rare diseases know no borders.

## Undiagnosed diseases network dataset

Here, we apply existing and newly developed statistical genetics methods to the Undiagnosed Diseases Network (UDN) cohort that includes extremely difficult-to-solve, likely genetic cases (Fig. 1a–e). The unique, diagnostically elusive presentation is the only criterion for inclusion, and patients have varied presentations including neurological, musculoskeletal, immune, endocrine, cardiac, and other disorders. Symptom onset ranges from neonatal through late adulthood. In contrast to most existing rare disease cohorts, individuals accepted to the UDN have already undergone lengthy but ultimately unfruitful diagnostic odysseys prior to enrollment. These patients subsequently undergo extensive phenotypic characterization at UDN clinical sites[15]. Both broad Human Phenotype Ontology (HPO) terms and highly detailed clinical notes are collected and made available for all UDN researchers. Phenotypic information includes laboratory evaluations, dysmorphology examinations, specialist assessments, surgical records, and imaging (Fig. 1f).

There is a similar emphasis on collecting sequencing data, with whole genomes sequenced for probands and their immediate or otherwise relevant family members. Although smaller than some other rare disease cohorts[14], the UDN—with a design bridging clinical, research and functional validation teams and a focus on extreme patient presentations—was thought to be optimized for case-based analyses where probands are evaluated individually in depth. Patients' detailed phenotypic information, ongoing confirmation of new diagnoses, and the potential enrichment for novel genetic disorders make for an ideal data space to validate and develop statistical approaches. We harmonized and jointly called single nucleotide (SNV) and insertion/deletion (indel) variants across 4,236 individuals with whole genome sequencing in the UDN dataset and additionally called de novo mutations from aligned reads across complete trios (Methods, Supplementary Fig. 1)[16].

## Clinical evaluation of computational findings

Here, we generate candidate gene–patient matches via a series of statistical genomic analyses implemented in our software suite, **Ra**re **Me**ndelian **Di**sease **E**nrichment **S**tatistics (RaMeDiES, Fig. 1g, h). We focus on the model of monogenic, autosomal inheritance in de novo and compound heterozygous cases to prioritize candidates via a genotype-first approach, with no clinical input or phenotypic information used. Each candidate is then evaluated with respect to the patient's clinical presentation and the gene and variant's putative role in disease—based on known disease associations, functionality in model organisms, tissue expression, molecular function, evolutionary constraint, and in silico predicted pathogenicity—to assess phenotypic match (Fig. 1i). For genes or gene pathways harboring deleterious variants across multiple individuals, phenotypic similarity between patients is also assessed. To scale clinical evaluation to the cohort level, we developed a semi-quantitative protocol guided by the ClinGen framework[17] that uses hierarchical decision models to increase efficiency and enables consistent and comparable evaluations of a gene–patient diagnostic fit by independent experts (Supplementary Note 2, Supplementary Fig. 3). We calibrated the protocol during development by testing whether the resulting clinical scores assigned by different experts on the clinical team were in agreement. We validated the protocol in a blind test using non-causative candidate genes as controls. Specifically, non-causative

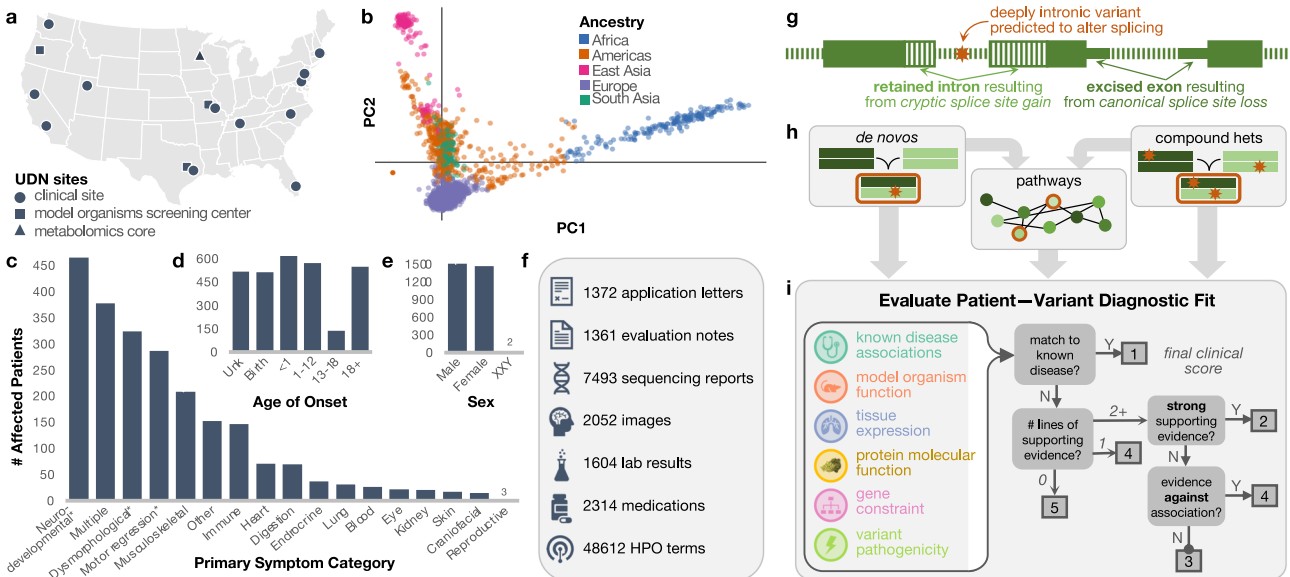

**Fig. 1 | Undiagnosed diseases network cohort analysis. a** Map of clinical and research sites within the Undiagnosed Diseases Network (UDN) for evaluating patients and candidate variant functionality. Map created using R's usmap package. **b** Genetic ancestry across the sequenced patient cohort. **c** Clinician-recorded primary symptom categories of patients. "Multiple" indicates 2+ categories could be considered primary and "other" indicates an unlisted category. Categories marked with an asterisk (*) are neurological subtypes (Supplementary Note 1). **d** Patient-reported age of first symptom onset. **e** Patient sex. **f** Categories and quantity of phenotype information collected for patients and made available to all UDN researchers. **g** Intronic variants detectable from genome sequencing (orange star) with a predicted splice-altering impact are considered alongside exonic variants in our statistical framework; these variants may result in retained introns or excised exons in processed transcripts. **h** We consider genes and gene pathways harboring de novo and compound heterozygous variants in sequenced trios (72% of all accepted cases). Other inheritance modes (e.g., homozygous, uniparental disomy) are not considered in our cohort-based framework. Complete case count by family structure (e.g., proband-only, duo) is in Supplementary Fig. 2. **i** Depiction of clinical framework to uniformly evaluate how well a patient's phenotypes are concordant with a candidate gene or variant. All icons in (**f**) and (**i**) are from Microsoft PowerPoint.

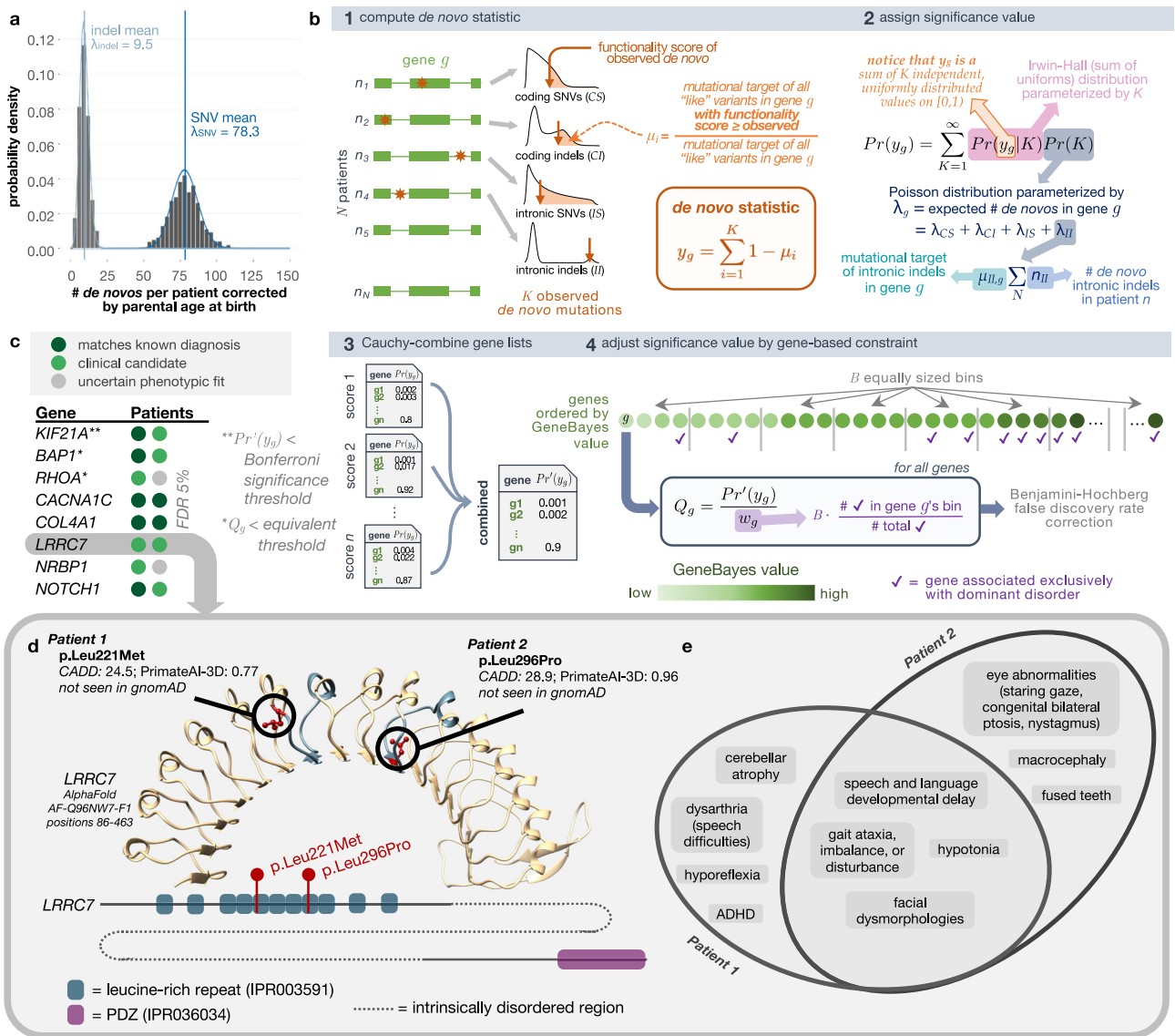

**Fig. 2 | De novo recurrence. a** De novo mutation counts per proband adjusted for parental ages. Blue vertical lines show the mean values of the distributions, and curves represent the Poisson fits. **b** Schematic of analytical test for the recurrence of *de novos* that considers distal splice-altering and exonic SNV and indel variants, their variant functionality scores, a genome-wide mutation rate model Roulette, and per-gene GeneBayes constraint values. "Like" variants refer to those of the same variant class (i.e., coding SNVs [*CS*], coding indels [*CI*], intronic SNVs [*IS*], intronic indels [*II*]) and within the same functionality score and minor allele frequency thresholds. **c** Genes with highest significance values for de novo recurrence across the cohort, computed as described in b3 ($Pr'(y_g)$) and b4 ($Q_g$), when focusing on missense variants with AlphaMissense and PrimateAI-3D scores; patients are

represented as colored circles. Complete gene list with exact *P* values can be found in Supplementary Data 3. Note that multiple testing corrections have been applied in the form of both Bonferroni (**) and FDR (*) thresholds. **d** AlphaFold-predicted human *LRRC7* protein structure (AF-Q96NW7-F1) covering the leucine-rich repeat region with high predicted structural confidence (amino acid positions 86-463). The fifth and eighth LRR domains where missense *de novos* were found are highlighted in blue. Reference alleles for missense de novo variants observed across two UDN patients (red) are shown in circles. A depiction of *LRRC7*'s linear protein sequence (Ensembl ID ENSP00000498937) with InterPro predicted domains shown in colored boxes is below. **e** Overlap of phenotype terms annotated to each patient.

genes were selected with identical criteria to true candidate genes, except biallelic variants were in *cis* rather than in *trans* or had low predicted pathogenicity scores. The clinical team applying the protocol consistently scored true candidate genes higher than control genes (Wilcoxon one-sided rank-sum *P* value = 0.0171, Methods, Supplementary Data 1), suggesting that the scores generated by the clinicians' protocol may be used to prioritize candidates.

## Results
### De novo analysis
Several highly penetrant, extreme phenotypic presentations underlying Mendelian and other congenital, complex human diseases have

been linked to de novo mutations[12,18,19]. We began by evaluating all independent, sporadic trios with complete sequencing data for de novo mutation etiologies. We detected 78.3 de novo point mutations and 9.5 de novo indels on average per proband genome concordant with the expectation[20]. Mutation count showed expected dependency on parental ages with Poisson-distributed adjusted counts, attesting to the quality of de novo calling (Fig. 2a, and Supplementary Data 2, Supplementary Fig. 4).

We then sought to identify genes enriched for deleterious de novo mutations across our patient cohort. Although the UDN dataset contains sequenced sporadic cases and their unaffected parents, classic case-control associated burden tests are underpowered in the de novo

regime because of the extreme rarity of these variants, difficulty in their identification without complete sequenced trios, and noise in variant sampling from both cases and controls[21]. Instead, de novo goodness-of-fit enrichment tests, which evaluate whether observed de novo events exceed what would be expected by chance based on mutation rates, are better suited for studying diseases with ultra-rare pathogenic variants[18,19]. The power of this de novo enrichment calculation increases with better models of underlying mutation rates and estimates of variant deleteriousness. Recently, the rate of de novo emergence has been estimated at basepair resolution with a high degree of accuracy[22]. Newly developed deep learning models for predicting the pathogenicity of de novo and other variants also now exhibit unprecedented accuracy in distinguishing disease-relevant variants[23,24]. We leverage these recent advances to build an accurate, unbiased statistical procedure called RaMeDiES-DN to detect genes enriched for deleterious *de novos*.

Unlike the earliest generation of de novo recurrence approaches which leveraged Poisson approximations for runtime efficiency but could not take advantage of improved deleteriousness scores and mutation rate models[18], RaMeDiES methods seamlessly incorporate per-variant deleteriousness scores and mutation rates without sacrificing runtime. Briefly, for a given observed variant in a gene, we define its "mutational target" as the sum of per-variant de novo mutation rates for all possible variants with as high or higher a deleteriousness score. By construction, this per-variant mutational target is expected to be a uniformly distributed statistic, enabling us to leverage closed-form analytical formulas for computing their cumulative probabilities (Supplementary Note 3). Our framework naturally combines different variant types including SNVs and indels with a distinct mutation rate model, and can interchangeably utilize various deleteriousness scores (Fig. 2b, Methods). Although current state-of-the-art de novo recurrence approaches also incorporate relevant variant-level information, they rely on a complex, permutation procedure[12]. RaMeDiES' analytical approach eliminates the need for permutation-based significance calculations and can process large datasets in mere seconds while maintaining well-calibrated *P* values (Supplementary Fig. 5).

We first focus on the subset of missense variants, which comprise a sizable proportion of known Mendelian disease-causing variants and for which new, specialized pathogenicity predictions exist (e.g., PrimateAI-3D and AlphaMissense)[23–25]. We find one significant gene, *KIF21A*, corresponding to the correct, complete diagnosis in one patient and a strong partial diagnosis in one other (Bonferroni-adjusted Cauchy-combined *P* value < 0.05, Fig. 2c). Notably, disease genes with a de novo mode of inheritance are expected to be under strong selection against heterozygous loss-of-function variants. We further refine our method to incorporate this intuition by prioritizing genes by their GeneBayes values, which indicate selection against heterozygous protein-truncating variants, using a weighted false discovery rate (FDR) procedure[26–28]. With this correction, we obtain three gene findings at an equivalent significance threshold (Q-value < 3e-6) and eight gene findings at FDR 5% (Supplementary Data 3). Our second and third gene hits, *BAP1* and *RHOA*, correspond to a known correct diagnosis in one patient and strong clinical matches in two other patients. Among the five remaining genes at FDR 5%, three genes (*CACNA1C, COL4A1* and *NOTCH1*) correspond to known diagnoses in five patients and the top clinical candidate in one patient. Two impacted patients with de novo missense variants in the leucine-rich repeat region of *LRRC7*, a gene expressed in the brain, had phenotypic overlap of hypotonia and developmental delay; one patient additionally experienced nystagmus, staring spells, and balance problems and the second had ataxic gait (Fig. 2d–e). We discussed these findings with the managing clinical teams overseeing these cases, and these UDN patients have now been included in a clinical case series linking *LRRC7* to an emerging neurodevelopmental disorder[13,29]. Another gene, *NRBP1*, remains a strong candidate in two patients due to their neurological phenotype

overlap and *NRBP1*'s expression in the brain. An initial functional study in fly through the UDN Model Organisms Screening Core was inconclusive. This gene has been submitted to MatchMaker Exchange.

We next consider all exonic variants, including nonsense variants and indels, and further incorporate additional well-established deleteriousness predictors, CADD and REVEL[30,31]. Different mutagenesis processes lead to indel mutations, so SNV mutation rate models can be inappropriate for modeling this mutation type for some genes[32]. We therefore constructed a separate per-gene mutation rate approximation for indels (see Methods for details). When we reran RaMeDiES-DN on all exonic variants using four deleteriousness predictors, we additionally identified *KMT2B* (Bonferroni-adjusted Cauchy-combined *p*-value < 0.05), corresponding to a correct diagnosis in four patients due to de novo indel variants (Supplementary Data 4, Supplementary Fig. 6a). The next seven gene findings at FDR 5% had already been identified when assessing recurrence of missense variants. At FDR 10%, we identify five new putative diagnoses. For instance, two patients had high impact missense de novo variants impacting *H4C5*, a histone gene that was not detected with significance in our missense-only enrichment test due to its lack of precomputed AlphaMissense scores. Both patients had infantile-onset gross motor developmental delays, dysmorphic facial features, and speech difficulties (Supplementary Fig. 6b,c). These and other phenotypes exhibited by each patient were recently found to be linked to missense variants in histone H4 genes[33]. For one of the patients, the de novo variant was contemporaneously interpreted by UDN clinical experts to be causal[34]. The second patient's de novo variant has now been reclassified as "pathogenic" and resulted in a new diagnosis for this participant. Two other patients with sporadic neurodevelopmental delay each harbor truncating de novo variants in *ZNF865*. Both patients have phenotypic overlap with a series of 10+ other patients with *ZNF865* mutations, which makes a compelling case for pathogenicity[35]. Subsequent to the publication of the case series, we anticipate this gene–disease relationship will be established as causal and both variants to be reclassified as likely pathogenic.

## Inclusion of deep intronic splice variants

Next, we demonstrate how RaMeDiES-DN can be extended to additionally consider non-exonic variants uncovered uniquely from whole genome sequencing using the same methodological infrastructure. On the one hand, it remains challenging to identify non-coding regulatory variants involved in rare Mendelian diseases[36], and the overall role of such variants in congenital disorders is still a subject of debate[37]. On the other hand, distal gain-of-splice site mutations creating new acceptor or donor splicing sites deep in the intronic sequences of genes are now a well-recognized cause of monogenic disease[38]. Identification of splice-altering variants directly from genome sequencing data is recently possible using newly-developed in silico predictive scores without relying on RNA sequencing. RNA sequencing has limitations for diagnosis because it depends on the availability of relevant tissue material that is especially challenging to obtain for neurodevelopmental patients, and it may miss lowly-expressed isoforms and those targeted by nonsense mediated decay[39]. Moreover, identifying disease-causal intronic splice variants is especially appealing due to their potential targetability using antisense oligonucleotide therapies[40].

Unlike functional predictions for exonic variants, which have been extensively validated for consistency and accuracy via decades of experimental in vitro and in vivo studies, functional predictions of splice-altering intronic variants are relatively new and still require experimental confirmation. We used a combined computational–experimental approach to prioritize distal splice variants using in silico predicted scores and an in vitro massively parallel splicing reporter assay (Methods, Supplementary Fig. 7)[41,42]. We found the per-variant in silico predictions to be mostly concordant with the in vitro assay readouts. Variants assigned higher in silico scores are

more frequently supported by the experimental, in vitro assay, and those with relatively lower in silico scores (SpliceAI <0.5) have a non-negligible validation rate as well (Supplementary Fig. 8). This prompted us to incorporate the full range of continuous SpliceAI scores, disregarding only the lowest scoring variants, in our statistics. We found this approach to consider distal splice-site variants attractive because it lends itself to a statistical analysis alongside exonic variants. Once genome-wide functionality score tracks are released for the next generation of splice predictors as well (e.g., Pangolin[43], they can be integrated into RaMeDiES using the same methodology leveraged for exonic variant predictors.

No new candidate genes with a significant recurrence of intronic *de novos* were found in the UDN dataset. However, by seamlessly incorporating non-exonic variants within the same statistical test, our approach enables a more complete, automated analysis of the growing volume of whole genome sequencing data across rare disease consortia.

We also ran the state-of-the-art de novo enrichment approach, DeNovoWEST[12]. Unlike our approach, DeNovoWEST incorporates a gain-of-function model alongside a loss-of-function model, which has the potential to yield additional findings. We equipped the DeNovoWEST algorithm with the Roulette mutation rate model, up-to-date CADD variant deleteriousness and $s_{het}$ gene constraint scores[27], and further incorporated deep intronic variants with predicted splice-altering impact (Supplementary Fig. 9). This approach yielded two Bonferroni-significant genes when applied to the UDN dataset, one of which was also uncovered by RaMeDiES-DN at Bonferroni significance and the second at a FDR of 6% (*KMT2B* and *H4C5*, Supplementary Fig. 10). We did not apply an FDR-based approach to DeNovoWEST's results to consider additional gene findings, because DeNovoWEST *P* values are a construct over three sometimes dependent tests, rendering an FDR adjustment inappropriate. We also find *CSMD1*, a highly indel-prone gene, within DeNovoWEST's top-ranked five genes, likely because indels and SNVs are not distinguished in the mutation rate model[44]. We then ran DeNovoWEST and RaMeDiES-DN on a larger cohort of previously published and jointly-analyzed de novo variants from the Deciphering Developmental Diseases (DDD) study, GeneDx, and Radboud University Medical Center (RUMC). Both methods recovered known autosomal dominant disease genes at a comparable rate (Supplementary Fig. 16, and Supplementary Data 5), but RaMeDiES-DN required less than one-tenth the runtime.

As opposed to methods like DeNovoWEST that require individual-level data, RaMeDiES' operation at the level of mutational targets enables sharing of intermediate statistics across cohorts without revealing patients' individual variants. The rare disease field has accumulated many individually small, but collectively large, cohorts that have never been included in any joint analyses because data governance restricts external sharing of variant- and patient-level data. As a proof of concept, we ran RaMeDiES-DN on the individual DDD, GeneDx, and RUMC de novo cohorts and subsequently combined the three summary-level mutational target statistics to generate gene findings. This meta-analysis run resulted in the same per-gene significance values as compared to RaMeDiES-DN's run on individual variants from the combined cohort (Supplementary Data 6).

## Compound heterozygous variant analysis

We next evaluate compound heterozygous (comphet) variants, which are the most likely cause of rare recessive disorders in populations with low degrees of consanguinity, as is largely the case in the United States[45]. Comphet variants are defined as a pair of alleles landing within the same gene and inherited in trans from unaffected parents who are also heterozygous at these loci. These inherited disease-causing variants tend to be rare in the population, due to the effect of selection against biallelic variant occurrences or against slightly deleterious phenotypes of heterozygous variants[46]. Despite the expected low

frequency of individual alleles comprising a comphet pair, directly selecting for highly deleterious comphet variants still results in numerous false positive findings at the cohort level, motivating a statistical approach for cohort-level comphet prioritization. Developing a statistical framework analogous to de novo goodness-of-fit enrichment requires modeling the distribution of rare inherited alleles per individual. De novo mutations arise through the universal process of mutagenesis and are therefore straightforward to model. Similarly, the distribution of the total number of all derived alleles per haploid genome (i.e., all non-ancestral variants inherited from one parent without any imposed frequency constraints) are also not dependent on the demographic history of the population and therefore are straightforward to model[47,48]. In contrast, however, the distribution of the total number of *rare* alleles per individual is highly dependent on population structure, which is notoriously difficult to account for. Some previous approaches for determining cohort-level significance of comphet variants ignore population structure when modeling the number of rare variants. Although this may be an accurate statistical test in controlled model organism cross experiments, it is inappropriate for natural human populations, where population structure is present even at a very fine scale[49]. In the Genome of the Netherlands (GoNL) dataset for instance, the number of synonymous singletons across unrelated individuals still reflects geographic structure along a south-north cline[50].

In our framework, we sidestep directly modeling the distribution of rare variant counts per individual and instead condition on the observed number of rare variants inherited from each parent using trio-level data. Given the number of rare variants inherited from each parent per individual, we then compute the probabilities of comphet variants landing in high-scoring positions in the same gene across the cohort. Although the positions where inherited variants land is influenced in part by direct and background selection and biased gene conversion, for very rare variants, the effect of these factors is negligible compared to the effect of the variation in mutation rate along the genome and the overall gene target size[22,51]. We therefore model the positional distribution of rare inherited variants using the same Roulette basepair-resolution de novo mutation rate model leveraged in our de novo recurrence model. Our comphet recurrence model, called RaMeDiES-CH, relies on the comphet mutational target, computed for each comphet variant pair and defined similarly as the de novo mutational target previously introduced. Specifically, the comphet mutational target is computed as the total *squared* mutation rate of all possible variants with higher functionality scores (Fig. 3a). RaMeDiES-CH applies the Cauchy *P* value combination approach as before to leverage multiple variant-level functionality scores while considering exonic and intronic variants, but does not incorporate gene constraint scores, which do not exist for recessive selection (Methods, Supplementary Fig. 11)[52]. RaMeDiES-CH computes well-calibrated per-gene *P* values for comphet variants in a cohort (Supplementary Fig. 5).

Across the set of non-consanguineous UDN families, we did not find significant recurrent comphet occurrences across genes. This result is unsurprising, as previous estimates suggest that in panmictic disease populations, only one deleterious comphet variant is expected for every five dominant *de novos*[49]. Nevertheless, RaMeDiES-CH represents an accurate and unbiased statistical test for the recurrence of comphet variants in human populations, which can be applied to reveal new diagnoses as sequenced rare disease datasets expand.

We suspected that singleton disease-causing comphet variants were still present in the cohort. We adapted our statistical framework to compute an individual-based statistic, RaMeDiES-IND, that normalizes each observed comphet variant mutational target across all genes in the genome rather than across all individuals in a cohort (Supplementary Fig. 12). This approach yielded a ranked list of patient–gene pairs across the UDN cohort, where each patient–gene pair could be annotated as corresponding to a correct diagnosis or

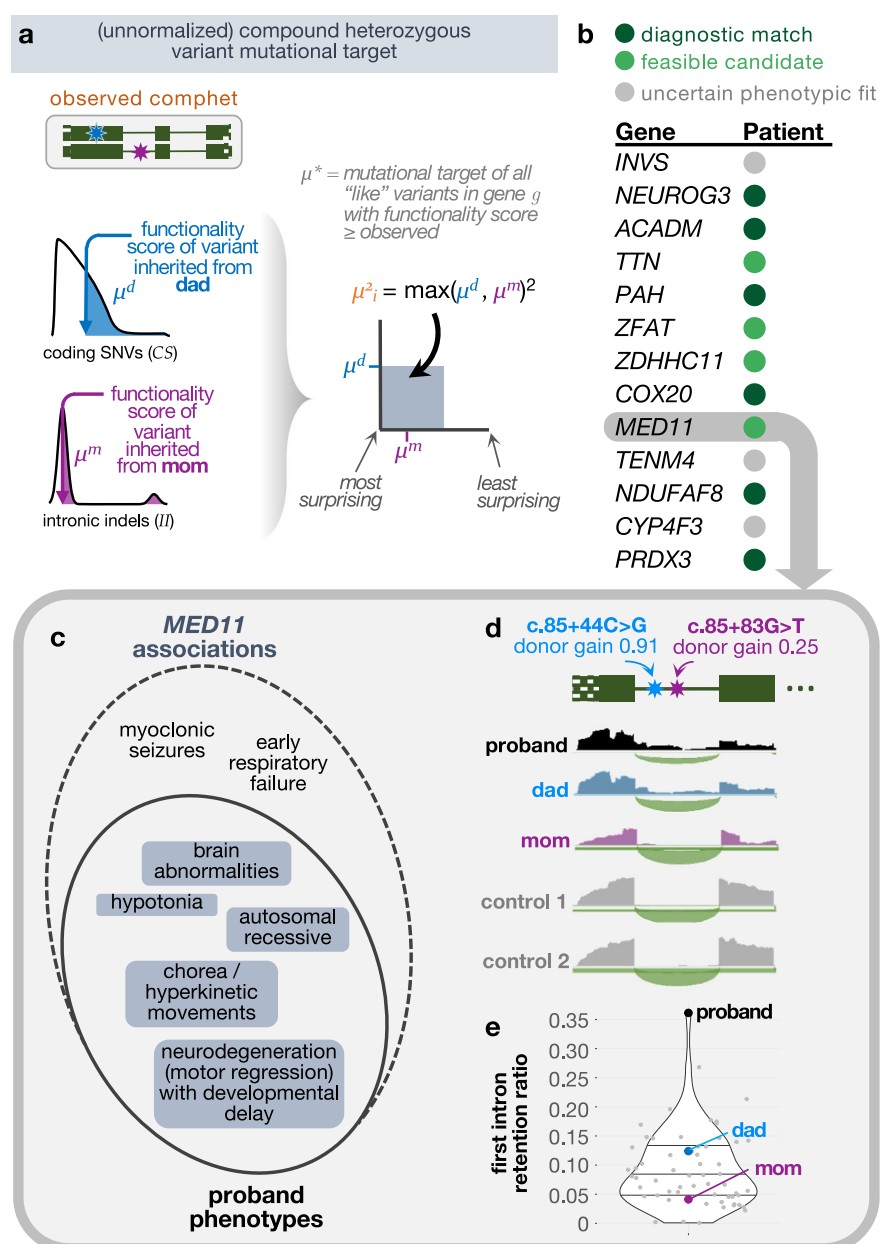

**Fig. 3 | Compound heterozygous variants. a** Illustration of the unnormalized squared mutational target computed for each observed comphet variant in a gene across the cohort (RaMeDiES-CH, Supplementary Fig. 11) or in an individual across the genome (RaMeDiES-IND, Supplementary Fig. 12). "Like" variants refer to those of the same variant class (i.e., coding SNVs [*CS*], coding indels [*CI*], intronic SNVs [*IS*], intronic indels [*II*]) and within the same functionality score and minor allele frequency thresholds. **b** Top ranked genes resulting in the best enrichment statistic computed for RaMeDiES-IND. Putative candidates refer to genes that remain candidates for pathogenicity due to their phenotypically-relevant tissue expression, but where there is not enough functional evidence or published gene–disease relationships to establish causality at this time. **c** Overlap between phenotypes associated with *MED11* and those exhibited by the affected patient. **d** RNA-Seq reads from whole blood samples aligned to first two exons and first intron of *MED11* for proband (black), dad (blue), mom (purple) and two tissue-matched control samples (gray). Thin green line represents the intron, solid boxes represent protein-coding exonic regions, and the dotted box represents the 5′ untranslated region of *MED11*. **(e)** Proband exhibits significant retention of the first intron relative to parents and fifty-three tissue-matched control samples. Intron retention ratio is calculated as the (median read depth of first intron) / (number of reads spanning first and second exons + median read depth of first intron).

---

otherwise (Supplementary Data 7). We computed a single enrichment statistic for this overall patient–gene ranking, which simultaneously suggested a threshold for clinical consideration of findings, as the best Fisher's exact test *P* achieved across all positions in the list. This enrichment statistic was significant when compared to the distribution of the same statistic computed across 10,000 random shuffles of the patient–gene list (permutation *P* value = 0.001, Methods, Supplementary Fig. 13). Among the top thirteen hits yielding this best enrichment statistic, we recapitulated five known diagnoses (i.e., *NEUROG3*, *PAH*, *COX20*, *NDUFAF8*, *PRDX3*)[53,54] and newly identified the genomic cause

of a known biochemical diagnosis (i.e., *ACADM* in a patient with MCAD deficiency). We also identified comphet variants in *MED11* which are now leading diagnostic candidates in an undiagnosed patient experiencing neurodegeneration, developmental delay, brain abnormalities, chorea, and hypotonia (Fig. 3c). *MED11* is associated with epilepsy and intellectual disability, and this patient's presentation could represent a phenotypic expansion of this known disorder[55]. Both inherited variants occur deep in the first intron of *MED11*, a region that would be missed by exome-only sequencing or analysis, and are predicted to cause cryptic splice donor gains. Transcriptome (RNA) sequencing of blood

samples from the affected patient and both parents highlighted a significantly higher rate of first intron retention in the affected patient relative to both parents and to fifty unrelated blood control samples (Fig. 3d–e, and Supplementary Fig. 14)[56].

Our comphet models assume that the two inherited alleles comprising a biallelic variant pair arose as a result of independent mutations. Homozygous recessive variants, where the exact same allele is inherited from each parent, are considered alongside compound heterozygous variant configurations in cases where the allele independence assumption is unlikely to be violated. In individuals with consanguineous parentage, however, the same heterozygous variant in both related parents most likely arose from a single, ancestral mutational event. As such, we directly exclude probands with evidence of parental relatedness from our analysis (Supplementary Note 4)[49].

### Pathway analysis

Genes involved in the same pathway are frequently involved in similar phenotypic presentations[57–60]. This provides an enticing possibility of drawing statistical power from multiple independent occurrences of deleterious variants in the same functional units, rather than just in the same genes. Moreover, therapeutics for disorders of the same functional unit that are individually too rare to meet minimal participant requirements for clinical trials may be evaluated together within the same umbrella or basket trial for more efficient approval[61]. However, such an approach should be pursued with caution, as the phenotypes stemming from perturbations of different genes in the same functional unit may vary to a great extent. Such differences in patient presentations may render the clinical evaluations and therapeutic potential of statistically significant findings virtually impossible. To mitigate this issue, we first initially consider groups of patients with similar phenotypes, and then within each of these groups, assess the over-representation of deleterious mutations across established biological pathways (Fig. 4a).

We start by clustering 2662 affected patients—with or without sequencing data—into 120 groups (median = 17, min = 2, max = 97 patients per cluster) based on the semantic similarity of their phenotype terms. Within each cluster, we then combine our de novo candidates, compound heterozygous candidates and known UDN diagnoses and perform gene set enrichment analysis (Methods, Supplementary Data 8). We focus our attention on undiagnosed cases with de novo or compound heterozygous candidates within enriched pathways in each cluster (Fig. 4b). We also report all enriched pathways including those with only diagnosed patients for potential therapeutic grouping (Supplementary Data 9).

Two of three total candidate genes in one cluster with 19 immunological disorder patients are both involved in the immunoproteasome complex (KEGG:03050, $n = 46$, adjusted $P$ value = 4.42e-3). One patient's genome contained a known diagnostic, de novo frameshift variant in *POMP*, an immunoproteasome chaperone protein[62]. An undiagnosed patient with evidence of chronic inflammation, recurrent infections, and skin lesions had a missense de novo in *PSMB8*, a component of the immunoproteasome β-ring with overlapping phenotypic associations (OMIM:256040). Both patients had similar combined immunodeficiency beyond what was captured in their standardized phenotype terms, including decreased global antibodies, decreased B cells and natural killer cells, and retained T cell functionality (Fig. 4c). Disruptions to immunoproteasome assembly and structure have been shown to lead to an accumulation of precursor intermediates, impaired proteolytic activity and subsequent uncontrolled inflammation[63].

In another cluster of 15 similarly presenting neurological patients, three candidate transmembrane genes were represented in the same functional pathway named for some genes' known involvement in taste transduction (KEGG:04742, n = 85, adjusted $P$ value = 7.45e-3). Two of these genes, *CACNA1C* and *GABRA3*, harbored high impact de novo and hemizygous missense variants respectively, corresponding

to known patient diagnoses[64,65]. The genome of another, undiagnosed, now deceased patient from this cluster with no prior candidate variants contained a synonymous de novo variant predicted to alter splicing in another gene in the same functional pathway, *HCN4* (Fig. 4d). All three patients exhibited seizures at a young age, speech delays, severe hypotonia, spasticity and visual impairment. Mouse knockouts of *HCN4* demonstrate neurological phenotypes[66,67]. In humans, *HCN4* is expressed in the visual and nervous systems and has recently been associated with infantile epilepsy, suggesting that this patient's undiagnosed disorder plausibly represents a phenotypic expansion of this gene[66,67].

### Discussion

In total, we analyze 886 sporadic or suspected recessive cases with complete trio or quad genome sequencing alongside an additional 463 phenotyped, diagnosed individuals using computational methods to identify de novo recurrence, compound heterozygosity, and pathway enrichment. We establish five new diagnoses and three new putative diagnoses in known disease-causing genes or genes previously unlinked to these patients' exact presentations. Our prioritization framework for pathway analysis further recapitulates 70 known de novo and 10 known comphet diagnoses and suggests 82 de novo and eight comphet candidates for follow-up (Methods, Supplementary Data 8).

In the field of common disease genetics, statistical inference of disease-associated genomic loci is confidently regarded as primary evidence for their causality. Rare disease genetics, in contrast, is in a transition state. Due to a lack of large disease-matched cohorts, case-based analyses of individual probands relying heavily on detailed patient phenotyping and clinical intuition have typically been used to generate candidate variant hypotheses. Evidence required to shift these variants from uncertain significance to known pathogenic status comes from experimental, functional studies and by identifying additional, unrelated, genotype-matched individuals with similar phenotypes through variant matchmaking services such as MatchMaker Exchange[68,69]. Recently, analyses of large, broadly-phenotyped cohorts of rare disease patients have demonstrated the potential for statistical approaches to reveal diagnoses and generate new gene discoveries in the rare disease space as well[12,14,70].

Although the genome is a big place, it is also a finite space with respect to gene regions impacted by simple variants such as SNVs and short (<10 basepairs) indels. This suggests that, in theory, recurrence-based statistical methods applied to sufficiently large sequenced cohorts of rare disease patients, even those with diverse phenotypic presentations like the UDN, will enable the eventual discovery of all causes of prenatally viable monogenic disease stemming from these variant types. In order to take statistical discoveries as primary evidence, as is the case for common diseases, we need accurate, well-calibrated statistical methods[71]. Even slight model misspecification may propagate and exacerbate the rate of false discoveries. The rapid growth of genomic datasets on which these models may be applied, coupled with an ongoing difficulty in phenotyping patients at scale to confirm findings[72], further increases the urgency for more rigorous models.

Here we show that well-calibrated statistical models can be built for both de novo and compound heterozygous modes of inheritance. Although novel disease–gene discovery from large, phenotypically- and genetically-homogenous cohorts has been demonstrated, we show here that rigorous analysis of a diverse, moderately-sized disease cohort at the gene and the pathway level shows promise.

Extending our RaMeDiES statistical framework to consider variant types with less accurate mutation rate models or pathogenicity predictions may lead to a sacrifice of precision. We therefore acknowledge the limitations of our statistical approach for comprehensive rare disease diagnosis efforts using short-read sequencing data, and suggest that our cohort-level models be applied alongside existing case-

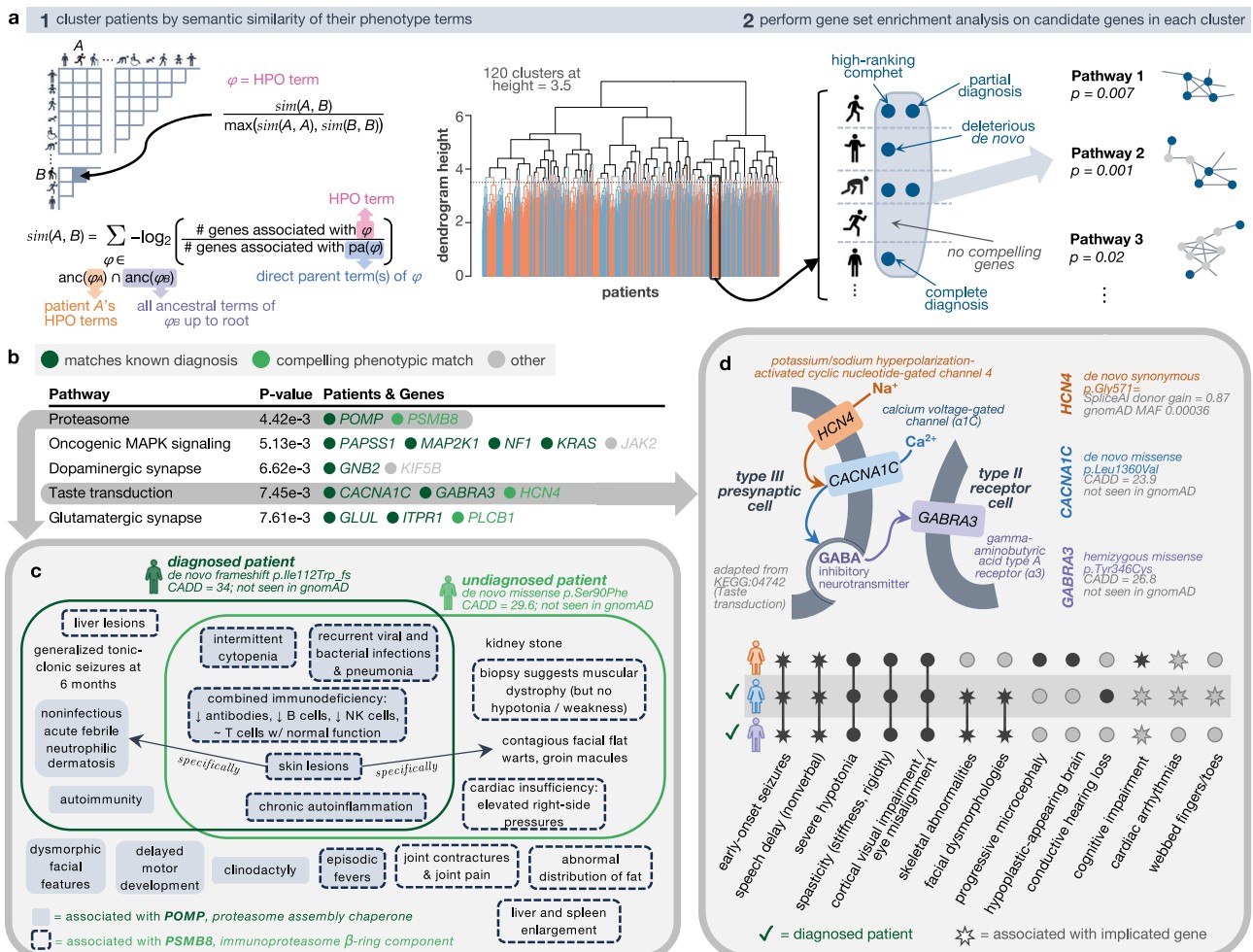

**Fig. 4 | Biological pathways enriched within phenotypically-similar patient subgroups. a** Schematic illustrating the two-step process of first clustering patients according to the semantic similarity of their phenotype terms and second finding enriched biological pathways among the genes within each patient cluster. **b** The most significant pathways per cluster (*P* value < 0.01, adjusted for multiple testing using g:Profiler's Statistical Correction Scheme[103]) with 1+ genes from 1+ undiagnosed patients; complete list in Supplementary Data 8. **c** Two patients with primarily immune-related symptoms each harbored a compelling de novo variant in genes involved in immunoproteasome assembly (*POMP*) and structure (*PSMB8*). Their symptoms strongly overlap, and a subset of these symptoms were also known to be associated with either gene in OMIM. **d** Three neurological patients had variants in transmembrane genes involved in the same pathway. These patients had substantial phenotypic overlap with each other, as expected, and with the phenotypes associated with each of their genes (depicted as star shapes in the upset plot). All icons in (a), (c) and (d) are from Microsoft PowerPoint.

based variant prioritization pipelines for additional statistically-meaningful findings.

First, although our models integrate non-coding variants with predicted splice-altering impacts, they do not consider potentially functional variants within whole genome data that fall into untranslated gene regions, RNA-coding genes or between genes, as genome-wide tracks of verifiable deleteriousness scores do not yet exist for these variant types. Improvements to and precomputed scores for these variants will be beneficial for interpretation efforts in general and can be leveraged in future iterations of RaMeDiES. Our statistical analysis also does not consider structural, large indel, copy number, or tandem repeat variants, as their identification from short-read sequencing data is computationally expensive and often inaccurate. Ongoing efforts to generate long-read sequencing data within the UDN and elsewhere should enable improved identification and analysis of pathogenic complex variants[73,74]. Developing a statistical model for these variants will still require accurate mutation rate estimates for these variant types, which is lacking. GnomAD-SV represents a promising iteration toward this goal, but is still highly dependent on their specific variant calling pipeline and data rather than biological mutagenic processes[75].

The presented method considers only autosomal de novo and compound heterozygous inheritance patterns due to complications in modeling other disease-relevant inheritance patterns. First, it is difficult to propose a statistical model for biallelic variant counts in consanguineous and founder populations, because these counts strongly depend on the ancestral population history and inbreeding patterns. A more appropriate statistical approach for assessing recurrence of these variants would be the extension of parametric linkage applied to very large cohorts[76]. Second, inclusion of hemizygous or other X-chromosome variants requires accurate sex-chromosome variant calling, which is notoriously error prone, as well as an accurate mutational model of the X chromosome, which is complicated due to sex-dependent selection and random X-inactivation[77]. Finally, although we do not model parental mosaicism or uniparental disomy in our recurrence statistics, these inheritance patterns and events are regularly assessed via complementary, traditional case-based approaches[10].

Even though genomic sequencing has been liberalized, currently many analyses are still restricted to individual programs, and regulatory and technical barriers prevent sharing individual-level variant data broadly. In contrast, there are avenues for sharing some variant-level data in a way that is easily accessible to clinical geneticists.

MatchMaker Exchange, for instance, enables the sharing of specific variants prioritized through case-based analyses with the goal of finding new genotype- and phenotype-matched patients. Broadening the success of MatchMaker Exchange to include variants that may not have risen to the level of strong candidates in analyses of individual patients is desirable. We developed a browser containing our gene-level findings and variant-level information about rare genetic variation in UDN patients (https://dbmi-bgm.github.io/udn-browser/). In addition, we provide an open-source software package, RaMeDiES, implementing the efficient and well-calibrated statistics for de novo recurrence and deleterious compound heterozygous inference proposed here. RaMeDiES' operation on shareable summary statistics rather than on variant-level data enables automated, deidentified cross-analysis of substantial existing yet siloed sequenced cohorts for new diagnostic discoveries. As the Mendelian genomics field continues the transition to this new data science phase, the methods we present here should facilitate the exciting prospect of international cross-cohort analyses, resulting in new findings and a vastly improved rare disease diagnostic rate globally.

## Methods

This work was performed in accordance with all ethical guidelines outlined in the NIH IRB #15HG0130. The study proposal and manuscript were approved by the UDN Publications and Research Committee. All participants have provided written, informed consent for the sharing and use of their data in this study.

### Undiagnosed diseases network (UDN) structure

The Undiagnosed Diseases Network (UDN) was established in 2014 with the goal of uncovering clinical diagnoses and novel disease-causing genetic variants and their molecular functionalities. In its current phase, the UDN is composed of 12 clinical research centers across the United States and a CLIA-certified sequencing core at Baylor Genetics. Typical UDN patients have already endured a multiyear "diagnostic odyssey" of extensive prior testing by multiple medical specialists and often inconclusive targeted, whole exome and even whole genome sequencing at the time of their application to the UDN.

As part of the application process, a team of clinicians and genetic counselors at one of the UDN clinical sites reviews the patient's medical records, referral letters and lab data and creates an abstracted case review document. If the team concludes that a UDN evaluation may aid in the identification of a diagnosis, the patient is accepted to the program and undergoes a thorough in-person evaluation at their assigned clinical site. Most patients and available affected and unaffected family members receive whole genome sequencing (GS) as well. All genomic sequencing data, clinical sequencing reports prepared in accordance with the American College of Medical Genetics and Genomics (ACMG) variant classification guidelines, structured phenotyping in the form of Human Phenotype Ontology (HPO) terms, lab results, imaging data, medication data, referral letters and clinical notes, the abstracted case review document, and candidate variants and genes are uploaded to the UDN Data Management and Coordinating Center. All patients enrolled in the UDN have consented to the broad sharing of all their genomic, phenotypic and clinical data with researchers network-wide for use in research projects and when evaluating gene–phenotype fit for a specific patient and candidate gene. Moreover, UDN patients have consented to follow-up if additional tests or information are deemed useful.

### Harmonization of whole genome sequencing data

Short-read whole genome sequencing was performed between 2014 and 2022 in accordance with the UDN Manual of Operations, which specifies that the average coverage across the genome must be >40X, and >97.5% of all coding and noncoding genes (UTRs, coding regions, and intronic regions) must be covered at >20X. Paired-end FASTQs

were retrieved in June 2022 for 4268 samples collected from 4236 unique individuals. Six individuals subsequently dropped out of the UDN program and are excluded from the analyses presented here. All FASTQ pairs were within expected parameters (https://www.bioinformatics.babraham.ac.uk/projects/fastqc/, version 08-01-19) and were aligned to human reference hg38 (with decoys and all alt contigs) using the Sentieon[78] bwa-mem implementation via the Clinical Genome Analysis Pipeline (CGAP, https://cgap.hms.harvard.edu/, version 29cefcec). Read groups were added via a custom CGAP script, multiple FASTQ pairs corresponding to the same sample were merged, and resulting BAMs were sorted, deduped, and recalibrated using a Sentieon implementation. GVCFs were produced using CGAP's implementation of GATK's HaplotypeCaller. All processing steps from FASTQ to GVCF were deployed on spot instances in Amazon Web Services (AWS). GVCFs were then egressed to the Harvard Medical School institutional cluster. SNVs/indels were jointly called across genomic shards then merged using Sentieon tools (version 202112.02). Per-sample sex and cross-sample relatedness were confirmed using Somalier (version 0.2.15, Supplementary Fig. 1)[79]. We required that all trios under consideration in our analysis had two parents reported as "unaffected", a child reported as "affected", parent–child relatedness $0.5 \pm 0.075$, parent–parent relatedness <0.15, mothers had heterozygous variants present and a scaled mean depth of ~2 on chromosome X, and fathers had a scaled mean depth of ~1 on chromosome Y. All variants were annotated using Ensembl VEP (version 108) and slivar (version 0.2.7) for TOPMed and per-population gnomAD (versions v2.1.1 and v3.1.2) variant frequencies and homozygote counts[80,81].

For our compound heterozygote analysis, we inferred within-family regions of identity by descent (IBD) using KING (version 2.3.0)[82]. We required at least one IBD region between the child and each parent to further confirm relatedness (in addition to kinship coefficient filtering) and no IBD regions of length > 3 Mb between parents to confirm non-consanguinity between parents. In families with multiple affected siblings, we select one sibling as the proband and disregard the other siblings during initial analyses. Variants in other affected siblings were then used to check segregation during validation of our findings. This process resulted in 846 non-consanguineous trios with an affected child and two unaffected parents for our analyses. We chose to stringently exclude individuals with evidence of familial consanguinity (i.e., by imposing a parental relatedness and IBD region length constraints) rather than excluding patients based on their relative recessive burden because an assumption of our statistical models is violated in consanguineous cases (Supplementary Note 4).

### Clinical evaluation framework

**Protocol overview.** We developed a clinical analysis protocol to reduce subjectivity in the assessment of diagnostic candidates. We used the case evaluation process implemented at Brigham Genomic Medicine as a foundation[83]. We then transformed this process into a systematic and structured protocol with inspiration from the gene–disease association criteria developed by the Clinical Genome Resource (ClinGen) group[17,68]. Evidence in support of or against a candidate variant–participant match was evaluated by a team of clinical geneticists according to three categories for experimental evidence not taken into account by our statistical analyses: (i) model organism or cell line studies, (ii) tissue expression, and (iii) protein molecular function. Clinicians also took into account case-level data and published literature with case-control data including (iv) known disease associations, (v) gene evolutionary constraint, and (vi) variant pathogenicity. Discrepancies in opinion were mediated by joint discussion until a consensus decision was reached. A detailed description of the protocol and scoring scheme is available in Supplementary Note 2 and hierarchical decision trees to streamline the scoring process are provided in Supplementary Fig. 2.

**Clinical score calibration.** We ensured that the protocol was specific and detailed to the extent that different clinicians with access to the same patient data would independently assign equal clinical scores to the same candidates. Over the course of two months, at least two clinicians each evaluated 2–3 compound heterozygous candidates per week and independently recorded their notes, final clinical scores, and score rationale in a REDCap database[84]. At weekly joint discussions, they iteratively updated the protocol to improve specificity and reduce discrepancies in scoring. The final two joint discussions confirmed that categorical and final clinical scores assigned by different clinicians were consistently in agreement.

**Validation.** The clinical team was provided with ten "candidates" and ten "decoys" from real UDN patients in random order to evaluate. The team was blinded to gene labels, variant inheritance and SpliceAI score information during evaluation. "Candidate" genes had two rare variants (gnomAD popmax AF < 0.001) inherited in *trans* where one variant was exonic with CADD > 23 and the second variant was intronic with a max SpliceAI > 0.3. "Decoy" genes were selected with identical criteria except that variants were actually inherited in *cis* or the intronic variant had a maximum SpliceAI score of 0. After assigning final clinical scores to each of the 20 genes, the candidate/decoy labels were revealed to the clinical team (Supplementary Data 1).

### Identification of de novo variants

For each of the 1463 sequenced trios in our harmonized UDN dataset, including trios with unaffected offspring, we select the subset of variants with read depth >10 and genotype quality (GQ) >20 across proband, mother and father. We further subset to variants with a "high" Roulette quality score, gnomAD population maximum allele frequency <0.01, TOPMed[85] allele frequency < 0.01, proband alternate allele read depth >4 and frequency > 0.2, and alternate read depth <2 in both parents.

We then utilize observed aligned reads across each trio and across thirty unrelated individuals to assign posterior probabilities to each putative de novo variant on autosomes using the CGAP reimplementation of novoCaller (https://cgap.hms.harvard.edu/)[16]. We consider all *de novos* with a novoCaller posterior probability >0.7 to be high confidence, noting that thresholding the novoCaller posterior probability from 0.5 to 0.95 has negligible impact on the number of passing variants overall and per-proband (Supplementary Fig. 6a). We further exclude probands with over 150 high confidence de novo calls, as these patients frequently had "suspected parental mosaicism" mentioned in their clinical records. Finally, because clonal sperm mosaicism may lead to siblings inheriting identical de novo variants, we exclude duplicate de novo variants within each family from downstream recurrence analyses[86]. This process resulted in 1072 trios with an affected proband and unaffected parents for further analysis.

### Analytical test for de novo cohort-level recurrence

**Basic statistic definition.** We define a cohort as a set of $N$ genomes (i.e., collections of genes) each with sets of de novo variants arising independently but based on the same background de novo mutation rate. Let $\mu_i$ denote the de novo mutation rate of a specific variant $i$. The mutational target of a gene $g$ is

$$\mu_g = \sum_{i \in g} \mu_i. \tag{1}$$

The mutational target of a variant $v$ in gene $g$ is

$$\mu_{g,v} = \sum_{i \in g} \mu_i \, \mathbb{1}_{score_i \geq score_v} \tag{2}$$

where $score_i$ is the deleteriousness score of variant $i$. Intuitively, the more surprising and/or deleterious a variant, the smaller its mutational target. By definition, variant mutational targets are uniformly distributed from 0 to $\mu_g$, so

$$\frac{\mu_{g,v}}{\mu_g} \sim U_{[0,1]} \tag{3}$$

ppose there are $K$ de novo variants falling within gene $g$ across the cohort, where $K \geq 1$. We define a statistic $y$ as

$$y = \sum_{i=1}^{K} \left(1 - \frac{\mu_{g,v_i}}{\mu_g}\right) \tag{4}$$

Note that $Y$ is a sum of $K$ uniformly distributed variables on [0,1] under the null. The distribution of $Y$ given parameter $K$ can thus be modeled by the Irwin-Hall (IH) "sum of uniforms" distribution, which has a closed form for its cumulative density function (CDF) and thus also for its survival function (SF), where $SF = 1 - CDF$[87]. This enables us to replace permutation-based significance evaluations and instead analytically compute the probability of achieving a $Y$ as high or higher than observed with $K$ variants using the IH survival function as $Pr(y|K) = IH_{SF}(y|K)$. We note that there are many other constructions over a set of uniformly-distributed random variables (such as $P$ values)[88,89]. We further note that as the cohort size dramatically increases, the Irwin-Hall distribution can be replaced with the normal distribution.

Finally, we also model $Pr(K)$, the probability of $K$ independent de novo variants to land in gene $g$ given this cohort of size $N$, to assign an overall significance value to our statistic $y$ as

$$Pr(y) = \sum_{K=1}^{\infty} Pr(y|K)Pr(K). \tag{5}$$

Because neither $y$ nor $Pr(K)$ are defined for $K = 0$, we do not expect $Pr(y)$ to be uniformly distributed. Instead, only $Pr(y|K \geq 1) = Pr(y)/Pr(K \geq 1)$ is expected to be uniformly distributed (Supplementary Fig. 9).

In a single genome with $n$ total observed de novo variants, the number of de novo variants to land in a particular gene $g$, given that $\mu_g \ll 1$, is Poisson distributed, parameterized by the expected number of de novos $\lambda = n\mu_g$. In a cohort of $N$ genomes, the number of de novo variants to land in gene $g$ is therefore a sum of $N$ Poisson-distributed random variables, which itself is also Poisson distributed. We thus compute $Pr(K) = Pois(K|\lambda)$, where $\lambda$ is given by

$$\mathbb{E}[K] \equiv \lambda = \mu_g \sum_{j=1}^{N} n_j. \tag{6}$$

### Different deleteriousness scores for coding and intronic variants

We use continuous, per-variant deleteriousness scores that are precomputed and publicly-available for all possible variants genome-wide in our computations. Precomputed scores are required for the calculation of comprehensive, basepair-resolution mutational targets as described above. For missense variants, we interchangeably use AlphaMissense (version hg38 released with their 2023 publication), PrimateAI-3D (academic license, accessed May 2024), CADD (version 1.6), and REVEL (accessed May 2024).[23,24,30,31] CADD is also used for scoring all other exonic variants, including nonsense and indel variants. For intronic variants, we use SpliceAI (academic license, accessed May 2021).[41] We use different variant functionality scores for exonic and intronic variants because we found that these values are poorly correlated with each other in intronic space (Supplementary Fig. 15). Clinical sequencing centers also regularly report these scores, suggesting their relevance in rare disorders[68,90].

## Different mutation rate models for SNV and indel variants

We use Roulette de novo mutation rates for SNVs genome-wide. Different mutational processes lead to indel mutations, so Roulette values cannot necessarily be adapted to model this mutation type.32 We approximated per-gene joint distributions of indel mutation rates and deleteriousness scores as follows. First, we considered all possible exonic indels of length ≤10nt for which precomputed CADD scores were available for download and all possible intronic insertions of length 1nt and deletions of length ≤4nt for which precomputed SpliceAI scores were available for download. Although SpliceAI provides predictions exhaustively for all possible indels, CADD provides scores for the subset of indels observed in gnomAD-v2. We excluded all indels that overlapped with any SNVs assigned a Roulette "low quality" filter, which are based on gnomAD quality metrics, abnormal density of segregating sites, and suspicious patterns of recurrence. We further excluded indels with a gnomAD popmax MAF >0.1% and/or a number of alleles in gnomAD (AN) in the bottom decile. For exonic and intronic variants separately, we binned all indels by their precomputed CADD or SpliceAI score rounded to the nearest hundredth. The total number of indels within a deleteriousness score bin and all bins corresponding to higher deleteriousness scores was used as an approximation to the mutational target associated with that score.

## Incorporation of different variant types

Because there are different deleteriousness scores for coding and intronic variants and different mutational targets for SNV and indel variants, we expand our basic test statistic to accommodate different variant types $t \in$ {coding SNV, coding indel, intronic SNV, intronic indel}. We redefine a gene and variant mutational target with respect to each variant type as

$$\mu_{g,t} = \sum_{i \in g,t} \mu_i \tag{7}$$

and

$$\mu_{g,v,t} = \sum_{i \in g,t} \mu_i \mathbb{1}_{score_i \geq score_v} \tag{8}$$

where $g,t$ refers to the subset of all possible variants in gene $g$ of type $t$. We define $y'$ as

$$y' = \sum_t \sum_{i=1}^{K} \left(1 - \frac{\mu_{g,v_i,t}}{\mu_{g,t}}\right) \sim IH\left(y' \middle| \sum_t K_t\right) \tag{9}$$

where $K_t$ is the number of observed de novo mutations of variant type $t$ landing in gene $g$. The expected number of de novos to land in gene $g$ when considering different variant types is

$$\mathbb{E}[K] \equiv \lambda' = \sum_t \mu_{g,t} \sum_{j=1}^{N} n_{j,t} \tag{10}$$

where $n_{j,t}$ denotes the total number of observed de novo variants of type $t$ in an individual $j$. For each variant type $t$, we scale $\mu_{g,t}$ such that $\sum_g \mu_{g,t} = 1$. We compute $Pr(K) = Pois(K|\lambda')$.

## Cauchy-combination of P values computed with different deleteriousness predictors

We can run our method using different deleteriousness score predictions for coding SNVs (i.e., AlphaMissense, PrimateAI-3D, CADD, or REVEL), resulting in slightly different lists of genes with corresponding P values when incorporating this variant type. We combine these lists using the Cauchy combination test, an analytic calculation that is applicable under arbitrary dependence structures[89].

## Incorporation of GeneBayes values

We incorporate GeneBayes values, which estimate the selection against heterozygous protein-truncating variants per gene, as weights in a weighted false discovery rate (FDR) procedure[26,91]. We sort all genes in ascending order by their GeneBayes values. We then separate these sorted genes into 10 equally sized decile bins. For each gene $g$ in each bin $b[g]$, we compute a weight $w_g$ as

$$w_g = 10 \cdot \frac{|DD \in b[g]|}{|DD|} \tag{11}$$

where $DD$ is the set of exclusively dominant disease-causing genes as annotated in OMIM (accessed December 2023). Genes without GeneBayes values are assigned a weight $w_g = 1$. Note that $E[w_g] = 1$ and that GeneBayes values, which are constant for all variants within a given gene, are independent from $y$ and $y'$ values, which vary for variants within a gene based on variant mutational targets and deleteriousness scores. This enables us to perform Benjamini-Hochberg false discovery rate correction on weighted Q-values computed for each $Pr(y')$ as $Q = Pr(y')/w_g$.26

## Massively parallel splicing reporter assay (MPSA)

**Assay design.** We designed oligonucleotides to evaluate the impact of a variant predicted to cause a cryptic splice site gain or a canonical splice site loss. For each variant with a predicted splice-altering impact, we extracted the surrounding genomic sequence from the UDN patient harboring the variant (alternate) as well as a paired version with the variant of interest replaced with the reference allele (reference). We centered the candidate sequence on the variant of interest, noting that the impacted splice site junction could be up to 50 nucleotides away from the variant. For a subset of variants, we also generated candidate sequences that were centered on the predicted site of the altered splice junction rather than on the variant itself. We embedded each candidate sequence in an oligonucleotide template containing a 4-nt barcode and flanking primers as follows:

## Splice donor library structure

GCACGGACAAAGTACTAGCC [155-nt candidate sequence][4-nt SD-associated barcode] GGAAGATCGACGCAGgtaagt

**Splice acceptor library structure.** TGCTCTTATGCGAACGTGTTAAC [4-nt SA-associated barcode] [151-nt candidate sequence] GGAAGATC GACGCAGgtaagtt

The final oligonucleotide library contained 6000 200-nt oligonucleotides, encompassing 1920 alternate/reference pairs, which we ordered from Twist Bioscience.

## Library cloning and experimental protocol

The oligonucleotide library was cloned separately using PCR amplification and NEBuilder assembly into lentiviral splice acceptor (pLenti-MPSA-acceptor) and splice donor (pLenti-MPSA-donor) vectors. These vectors consisted of an EF1A promoter and an mCherry open reading frame (ORF) followed by splicing reporter modules based off of prior published massively parallel splicing reporter constructs[92,93] (Supplementary Fig. 7) as well as a separate Puromycin selection cassette. Plasmids have been deposited to Addgene under accession numbers 240805 ("plentiMPSA SAentry PuroR") for splice acceptors and 240806 ("plentiMPSA SDentry PuroR") for splice donors.

Lentiviral particles for each library were produced and titrated. Each library was transduced at a multiplicity of infection (MOI) of 0.3 in three biological replicates into 6.25*106 cells/replicate of HepG2 (liver cell line derived from primary cells extracted from a white, male, 15-year-old with liver cancer, catalog #HB-8065) and SK-N-SH (neural-like cell line derived from primary cells extracted from a female 4-year-old, catalog #HTB-11), both acquired from American Type Culture

Collection (ATCC). Cells were routinely tested for mycoplasma contamination via qPCR, and all tests were consistently negative for mycoplasma. Cells were selected with Puromycin to completion, and genomic DNA and RNA were harvested one week after transduction.

PCR-based nextgen sequencing (NGS) library preparation was performed on all 12 genomic DNA and RNA samples. Libraries were sequenced with 75-nt paired-end reads using an Illumina NextSeq 500 sequencer, ensuring an average of >1000 reads per library member from all libraries.

## Barcode mapping

Over ~75% of all RNA reads could be mapped back to a 15-nt barcode found in our starting dictionary. This resulted in ~6−15 million mapped RNA reads per MPSA replicate, yielding a median of 1170 mapped reads per alternate/reference library pair per replicate. Results from Tapestation, an automated electrophoresis system for sizing and quantifying nucleic acid samples, showed that 49.6% of mapped reads from splice donor MPSA experiments utilized some library splice donor site and 50.4% utilized the experimentally fixed site. Across splice acceptor MPSA experiments, 58.3% of mapped reads utilized some library splice acceptor site and 41.7% utilized the fixed site.

## MPSA validation rate

We considered all alternate/reference library pairs with at least 10 barcode-disambiguated mapped reads each in one or more MPSA experiments; 99.4% of pairs met this requirement. Each read was then categorized as (1) using the experimentally fixed splice site, (2) using a splice site corresponding to a known intron/exon junction as annotated in Ensembl, (3) using the SpliceAI-predicted cryptic splice gain site, (4) using a cryptic splice site at a different location, (5) malformed where the read did not begin with the correct fixed sequence due to a next-generation sequencing error, or (6) recombined where the read did not align to the expected oligo sequence at all. The percent of malformed and recombined reads per alternate/reference pair was 7.5% (SD=1.9%) and 6.2% (SD=10.6%) respectively on average. The position of SpliceAI-predicted cryptic splice sites often did not correspond to the expected splice junction based on manual inspection or to the splice sites observed in MPSA experiments (55.4% of splice acceptor and 5.4% of splice donor predicted positions matched). We instead considered the most common cryptic splice site position observed in each alternate library sequence to be the predicted site. MPSA validation rate is computed per alternate/reference library pair as the difference in percentages of total reads supporting the predicted cryptic splice site between oligos containing the alternate variant and the corresponding reference oligonucleotides (Supplementary Fig. 8a).

We compared the MPSA validation rates across the three biological replicates and two cell types using Pearson's correlation (Supplementary Fig. 8b).

## DeNovoWEST gene-specific enrichment of de novo variants

We modified the DeNovoWEST weighted permutation test by first augmenting the set of variants under consideration beyond exonic variants to include all possible intronic variants in protein-coding genes with a SpliceAI score >0.4, resulting in ~400k additional possible variants under consideration.[41] To this end, we modified the codebase (version v1.0.0) to consider these intronic putatively splice-altering variants to have the same functional consequence as canonical splice site variants if they had a VEP annotation of "splice_acceptor" or "splice_donor" or the same functional consequence as missense variants otherwise. We then updated the required precomputed values, including per-variant mutation rates, minor allele frequencies, deleteriousness scores and per-region constraint values as detailed below, for all exonic and intronic variants under consideration (Supplementary Fig. 9a). The underlying triplet-context mutational model was

replaced with genome-wide, per-SNV Roulette mutation rate estimates[94]. Each variant's minor allele frequency was set to the maximum gnomAD-v3 population or TOPMed allele frequency. Per-variant Phred-scaled and unscaled CADD values were obtained from (https://cadd.gs.washington.edu/) (version 1.6 for GRCh38/hg38). Updated per-gene shet values were obtained from (http://genetics.bwh.harvard.edu/genescores/selection.html) and binned into a "low" category if mean shet was below 0.15 and a "high" category otherwise.27 Notably, some stable Ensembl gene IDs in GRCh37/hg19 are not present in GRCh38/hg38 and vice versa; all variants from the 894 GRCh38/hg38 genes without shet values are binned into the "low" category. Regional missense constraint values, defined for adjacent windows covering the full genomic region of each protein-coding gene were obtained from (https://gnomad.broadinstitute.org/downloads#exac-regional-missense-constraint). We translated these genomic region coordinates from GRCh37/hg19 to GRCh38/hg38 using UCSC's LiftOver tool and then assigned a constraint value to exonic and intronic variants corresponding to the genomic region they fell into. We recomputed the weights assigned to each variant type using the union of all de novo variants in our cohort and the de novo variants released with DeNovoWEST (encompassing ~31,000 exome-only trios), because the distribution of de novo variant classes in UDN data was very similar to the distribution of de novo variant classes in the dataset used by DeNovoWEST (Supplementary Fig. 9b-c) and because the authors warn that weights generated from smaller datasets alone may be unreliable. Gene severity scores were then computed for every gene harboring one or more de novo variants across our cohort. We adjust DeNovoWEST assigned p-values using Bonferroni correction for twice the total number of genes evaluated as suggested by the authors. We find that DeNovoWEST and RaMeDiES-DN (using only CADD in exonic regions as a closer comparison to DeNovoWEST) recovered known autosomal dominant disease genes at a comparable rate across de novo variants provided in the original DeNovoWEST paper (Supplementary Fig. 16).

## Analytical test for compound heterozygous cohort-level recurrence

A compound heterozygous configuration is an independent occurrence of two variants: one maternally ($M$) and the other paternally ($D$) inherited. Here we use the term "compound heterozygous configuration" to refer to either two distinct heterozygous variants inherited from either parent, or a single homozygous recessive variant where both independently-inherited alleles arose from independent mutational events (Supplementary Note 4). The mutational target of a compound heterozygous configuration should therefore lie in a space of squared mutational targets. We define the mutational target of a compound heterozygous configuration as

$$\mu_{g,v_M,v_D} = \max\left(\mu_{g,v_M}, \mu_{g,v_D}\right)^2 \tag{12}$$

where $v_M$ and $v_D$ are maternally and paternally inherited variants comprising a compound heterozygous configuration, and $\mu_{g,v_M}$ and $\mu_{g,v_D}$ are computed as in Eq. 12. To prioritize compound heterozygous configurations with both deleterious variants, we use the maximum over per-variant mutational targets. A single deleterious variant in a compound heterozygous configuration may indicate carrier status rather than a compelling candidate for a rare disorder. By this definition, $\mu_{g,v_M,v_D}$ is uniformly distributed at null (Supplementary Note 4). This enables us to define a similarly constructed statistic $y^c$ modelable by the Irwin-Hall distribution as in the case of recurrent de novos (Eq. 4):

$$y^c = \sum_{j=1}^{K} \left(1 - \frac{\mu_{g,v_{M,i},v_{D,i}}}{\mu_g^2}\right) \sim IH(y^c|K) \tag{13}$$

where $K$ is the number of compound heterozygous configurations independently landing in gene $g$ across the cohort, and $v_{M,j}$ and $v_{D,j}$ are the maternally and paternally inherited variants in gene $g$ in individual $j$. As before, $K$ is approximately Poisson distributed, and parameter $\lambda^c$, the expected number of compound heterozygous configurations to land in gene $g$, is given by

$$\mathbb{E}[K] \equiv \lambda^c = \mu_g{}^2 \sum_{j=1}^{N} n_{M,j} n_{D,j} \qquad (14)$$

where $n_{M,j}$ and $n_{D,j}$ are the numbers of maternally and paternally inherited rare variants in an individual $j$, respectively. We compute $Pr(K) = Pois(K|\lambda^c)$ as before.

Finally, we extend this basic test statistic to accommodate 16 compound heterozygous configuration types as $(t_M, t_D) \in \{\text{coding SNV, coding indel, intronic SNV, intronic indel}\}^2$ and define $y^{c'}$ and Poisson parameter $\lambda^{c'}$ accordingly as

$$y^{c'} = \sum_{t_M, t_D} \sum_{i=1}^{K_{t_M, t_D}} \left( 1 - \frac{\mu_{g, v_{M,i}, t_M, v_{D,i}, t_D}}{\max\left(\mu_{g, t_M}, \mu_{g, t_D}\right)^2} \right) \qquad (15)$$

and

$$\mathbb{E}[K] \equiv \lambda^{c'} = \sum_{t_M, t_D} \mu_{g, t_M} \mu_{g, t_D} \sum_{j=1}^{N} n_{M, t_M, j} n_{D, t_D, j} \qquad (16)$$

where $K_{t_M, t_D}$ is the number of compound heterozygous configurations in gene $g$ across the cohort where the maternally inherited variant is of type $t_M$ and the paternally inherited variant is of type $t_D$. Instances where $K_{t_M, t_D} = 0$ are excluded from the above sums. For each variant type $t$, we scale $\mu_{g,t}$ such that $\sum_g \mu_{g,t} = 1$. We compute the probability of $y^{c'}$ as in Eq. 5. Note that inherited homozygous recessive variants that arose from a single, ancestral mutational event violate the assumptions of our approach and are excluded (Supplementary Note 4).

## Modeling false positive diagnoses

For any gene where the observed number of variants $K > \mathbb{E}[K]$ across the cohort, we suspect that there are some true diagnoses in specific patients as well as some "false positives" where a randomly occurring variant in a patient is unrelated to the patient's condition. We use the binomial distribution parameterized by $K$ independent trials and probability of success per trial $\mathbb{E}[K]/N$ to estimate the proportion of false positive diagnoses for each gene.

## Analytical test for individual-level compound heterozygous configuration

Given a set of independent compound heterozygous configurations across genes in a single individual's genome, we construct a test for the hypothesis of a monogenic, recessive disorder caused by one of these compound heterozygous configurations against the null. We assume up to one compound heterozygous configuration per gene, i.e., for each gene $g$, $n_M n_D \mu_g{}^2 \ll 1$, where $\sum_g \mu_g = 1$ and $n_M$ and $n_D$ are the numbers of maternally and paternally inherited rare variants in this individual's genome.

We now rescale the mutational target of a compound heterozygous configuration (Eq. 12) with respect to all genes $g$ in the genome as

$$\bar{\mu}_{g, v_M, v_D} = \frac{\sum_{i \in G} min\left(\mu_i{}^2, \mu_{g, v_M, v_D}\right)}{\sum_{i \in G} \mu_i{}^2}. \qquad (17)$$

Intuitively, this corresponds to the probability of observing a compound heterozygous configuration with an equal or smaller (i.e.,

more surprising) mutational target occurring in any gene in the genome. Thus, $\tilde{\mu} \sim U_{[0,1]}$. We precompute each gene's compound heterozygous mutational target $\mu_i{}^2$ for all genes in the genome in order to quickly compute $\tilde{\mu}$ values for each observed compound heterozygous configuration in an individual.

Next, we define our statistic $\tilde{y}^c$ per individual as the minimal observed rescaled compound heterozygous mutational target:

$$\tilde{y}^c = min(\tilde{\mu}_1, \dots, \tilde{\mu}_K) \qquad (18)$$

We compute the probability of observing a $\tilde{y}^c$ value this low or lower given $K$ total genes with observed compound heterozygous configurations in an individual's genome as

$$Pr(\tilde{y}^c | K) = 1 - \prod_{i=1}^{K} Pr(Y > \tilde{y}^c) = 1 - Pr(Y > \tilde{y}^c)^K. \qquad (19)$$

where $Y$ is a dummy variable. Because $\tilde{y}^c$ is uniformly distributed on $[0,1]$, $Pr(Y > \tilde{y}^c) = (1 - \tilde{y}^c)$, so we simplify this calculation as

$$Pr(\tilde{y}^c | K) = 1 - (1 - \tilde{y}^c)^K \qquad (20)$$

We also model the distribution of $K$ observed compound heterozygous configurations across an individual's genome in order to compute the overall probability of our statistic $\tilde{y}^c$ using the same formulation as before (Eq. 5). The distribution of $K$, given our prior assumption of at most one compound heterozygous configuration per gene, has an exact solution as the number of double events in a bivariate binomial distribution with correlation parameter $\rho$ capturing the effect of different gene lengths on $K$. However, due to the complexity in calculations of the exact solution, here we use the Poisson approximation instead because, for each gene $g$, $\sum_g \mu_g = 1$ and $n_M n_D \mu_g{}^2 \ll 1$. The $\tilde{\lambda}^c$ parameter for the Poisson approximation in this case is

$$\mathbb{E}[K] \equiv \tilde{\lambda}^c = n_M n_D \sum_{i=1}^{G} \mu_i{}^2 \qquad (21)$$

Finally, we accommodate the 16 compound heterozygous configuration types as $(t_M, t_D) \in \{\text{coding SNV, coding indel, intronic SNV, intronic indel}\}^2$ and redefine $\tilde{\mu}'$, $\tilde{y}^{c'}$ and Poisson parameter $\tilde{\lambda}^{c'}$ accordingly as

$$\tilde{\mu}'_{g, v_M, v_D} = \frac{\sum_{t_M, t_D} \sum_{i \in G} min\left(\mu_{i, t_M} \mu_{i, t_D}, \mu_{g, v_{M, t_M}, v_{D, t_D}}\right)}{\sum_{t_M, t_D} \sum_{i \in G} \mu_{i, t_M} \mu_{i, t_D}} \qquad (22)$$

and

$$\tilde{y}^{c'} = min(\tilde{\mu}'_1, \dots, \tilde{\mu}'_K) \qquad (23)$$

and

$$\mathbb{E}[K] \equiv \tilde{\lambda}^{c'} = \sum_{t_M, t_D} n_{M, t_M} n_{D, t_D} \sum_{i \in G} \mu_{i, t_M} \mu_{i, t_D} \qquad (24)$$

## Enrichment for correct diagnoses

Given a ranked list of genes across a cohort of patients, where each gene may be diagnostic for the given patient, we can compute enrichment for correct diagnoses at each gene rank. We use Fisher's exact test to compare the proportion of complete, certain diagnoses in all genes up to and including rank $k$ compared to the proportion of correct diagnoses at genes ranked $k$+1 through the end of the list. We consider the minimum Fisher's exact test P across all $k$ to be our overall enrichment. We assign a

permutation-based $P$ value to this enrichment value by randomly permuting the initial gene list 10,000 times and recomputing the minimum Fisher's exact test P for each permuted list.

## Transcriptome sequencing analysis for *MED11*

**RNA extraction, sequencing and quality control.** RNA was extracted from UDN patients' whole blood samples received at UCLA between 2018 and 2019 using PAXgene Blood RNA extraction kits from Qiagen. Concentration of RNA in each sample was quantified using the Qubit 3.0 Fluorometer. RNA integrity numbers (RINs), a quality control measure, were assessed per sample using the Agilent bioanalyzer. RNA libraries were prepared for each sample using either the NuGEN Universal Plus mRNA-Seq kit or the Illumina TruSeq mRNA + Globin Minus kit. Sequencing was then performed on the Illumina NovaSeq 6000 to generate ~50-100 million 100-150bp paired-end reads per sample. Library preparation and sequencing were performed at the UCLA Neuroscience Genomics Core and the UCLA Technology Center for Genomics and Bioinformatics Core. Sequenced reads in FASTQ format were aligned to human reference genome GRCh37 using STAR v2.5.2b with default parameters and Gencode v19 annotations[95,96]. To increase sensitivity to novel splice junctions, reads were mapped using the STAR 2-pass mode, where novel splice junctions detected during the first pass alignment are indexed and used alongside known splice junctions in the second pass remapping. We confirmed effective ribosomal RNA (rRNA) depletion per sample by aligning all paired-end reads to the complete sequences for nuclear and mitochondrial rRNAs using BWA-mem v0.7.17[97] and ensured that the proportion of aligned reads did not suggest excessive rRNA contamination. Duplicate reads were marked using PicardTools v4.2.4.0 and post-alignment sequencing quality was assessed using RNA-SeQC v1.1.8 to ensure adequate library complexity[98,99]. RNA sample identity was confirmed by comparing single nucleotide variant (SNV) calls from RNA sequencing to SNV calls from corresponding exome or genome sequencing data per sample.

## Intron retention outlier analysis

Fifty-three tissue-matched control samples from UDN participants unrelated to the proband, mother and father were selected for outlier analysis. IRFinder v1.2.456 was run in BAM mode using the same human reference GRCh37 and Gencode v19 annotations on aligned BAM files to measure the level of intron retention (i.e., "IRratio") in *MED11* across the proband, mother, father, and control samples. The IRratio is computed per sample as (median read depth of first intron) / (number of reads spanning first and second exons + median read depth of first intron). Aligned reads covering the *MED11* gene region (i.e., chr17:4,634,723-4,636,903) from the proband, mother, father and two control samples were viewed using a local installation of the Integrative Genomics Viewer (IGV) v2.16.0[100].

## Pathway enrichment analysis

**Phenotypically-similar patient groupings.** Phrank (version v2018-12-13) was used to compute all-against-all pairwise phenotype similarity scores between all affected patients' sets of standardized HPO terms. We normalized these scores by dividing by the maximum self-similarity score in each pair[101]. UDN patients experience a spectrum of symptoms across overlapping biological categories and therefore cannot be easily separated into distinct, well-defined clusters (Supplementary Fig. 17). We iteratively grouped similar patient pairs using complete-linkage hierarchical clustering with the agnes function from R's cluster package (version v2.1.4), which allows for patient groups of different sizes while minimizing the maximum distance between any two patients in the same cluster. We assigned patients to clusters by cutting the resulting dendrogram at height=3.5, resulting in 120 clusters of 2–97 patients per cluster (mean=22, median=17).

## Selecting genes per patient cluster

We identify genes per patient cluster as follows. First, we consider known diagnoses for all UDN patients in that cluster. For patients with a diagnosis that was "complete" (i.e., explained all symptoms including asserted phenotypes), no further genes are considered. For patients with no diagnosis or at most one "partial" diagnosis, we then consider genes with an exonic (or intronic with a SpliceAI score >0.4) de novo variant and assign each gene its variant's severity weight ($s$) from our modified DeNovoWEST procedure. Recall that weights are assigned per variant class based on functional impact (e.g., frameshift, nonsense, missense), variant deleteriousness, and gene constraint. Autosomal de novos are considered as before in addition to de novos on chromosome X with gnomAD population maximum allele frequency < 0.0001, TOPMed allele frequency <0.0001, proband alternate allele read depth >20 and frequency >0.2 (for females) and alternate read depth >20 in both parents. We consider the most significant gene per patient with $Pr(s) < 0.0005$, where $Pr(s) = Pr(S \geq s | K = 1) Pr(K = 1)$. $Pr(S \geq s | K = 1)$ is computed exactly using precomputed per-variant Roulette mutation rates and variant weights per gene. $Pr(K=1)$ is computed assuming mutations follow a Poisson distribution with $\lambda = \mu_{gene}$ for genes falling on chromosome X in males and $\lambda = 2\mu_{gene}$ otherwise. Finally, in patients who still have fewer than two genes at this point and no complete diagnoses, we include up to one additional gene harboring a compound heterozygous variant pair that ranked in the top 100 in our RaMeDiES-IND cohort-wide per-individual analysis, as there was significant enrichment for correct diagnoses in this set (Fisher's exact test $P$ = 5.23e-4). Across all patient clusters, we considered 70 genes (6.15%) with de novo variants corresponding to known diagnoses, 10 genes (0.88%) with compound heterozygous variants corresponding to known diagnoses, 562 (49.38%) other known diagnostic genes, 434 genes (38.14%) with new de novo candidates, and 62 genes (5.45%) with new compound heterozygous candidates.

## Gene set enrichment analysis (GSEA)

The genes found across all patients in each patient cluster were used as a query set for gene set enrichment analysis (GSEA) using g:Profiler (version e108_eg55_p17)[102]. We considered Reactome and KEGG biological pathway gene sets of size <150 genes and set our background gene set to all human genes annotated in Ensembl. Enrichment $P$ values are adjusted using g:Profiler's g:SCS approach[103]. Briefly, for every query gene set size, 2000 random gene sets of the same size are used as queries for GSEA with the same parameters, and the lowest pathway enrichment $P$ value is recorded for each random query set. A threshold $t$ is selected for each query gene set size as the 5% quantile of these random minimum $P$ values. Enrichment $P$ values resulting from the true gene query are then adjusted by multiplying by 0.05/$t$.

## Reporting summary

Further information on research design is available in the Nature Portfolio Reporting Summary linked to this article.

## Data availability

Data was collected centrally by the UDN Data Management and Coordinating Center (independently of this work) as described in the Manual of Operations (https://undiagnosed.hms.harvard.edu/research/udn-manual-of-operations/). The deidentified genome data, transcriptome data, and corresponding phenotype data in the form of HPO terms used in this study have been deposited in the dbGaP database under accession phs001232.v5.p2 (https://www.ncbi.nlm.nih.gov/projects/gap/cgi-bin/study.cgi?study_id=phs001232.v5.p2). Genome-wide, rare SNV and indel variants and HPO codes for UDN participants included in this study are queryable in our public-facing

browser (https://dbmi-bgm.github.io/udn-browser/). Standardized phenotype data and candidate genes and variants used in this study have been submitted to the Matchmaker Exchange database (https://www.matchmakerexchange.org/). Variant-level data, clinical significance and supporting evidence, demographic information, and phenotype information for all candidate and diagnostic variants, including those identified through this study, have been submitted to the ClinVar database (https://www.ncbi.nlm.nih.gov/clinvar/). Identifiable patient data is available only to UDN investigators under restricted access to protect patient privacy in compliance with patient consent. Other relevant, deidentified patient-specific clinical information available to researchers involved in this study, but not available publicly, may be shared on a case-by-case basis at the discretion of the corresponding clinical team if it is directly related to diagnosing or potentially treating the patient. Requests for such data can be directed to udncc@hms.harvard.edu with a response expected within six weeks. There are no specific reporting restrictions on data use outlined within the UDN centralized Data Use Agreement. However, each query will be reviewed by the respective institution to verify if the request is subject to any intellectual property or confidentiality obligations. Plasmids used in the MPSA experiment have been deposited to Addgene under accession numbers 240805 ("plentiMPSA SAentry PuroR") and 240806 ("plentiMPSA SDentry PuroR") (https://www.addgene.org/).

## Code availability

Our software package is available at https://github.com/hms-dbmi/RaMeDiES[104].

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

## Acknowledgements

The authors would like to thank following individuals and organizations: Feruza Abraamyan for her contribution in the initial stages of developing the clinical evaluation protocol, Tian Yu for formatting RNA library reads for the MPSA analysis, the Undiagnosed Diseases Network Tool Building Coalition working group for advice on variant calling and sequencing quality metrics, Cecilia Esteves for sequencing file management, Amazon Web Services for complimentary data processing cloud credits, Rafael Aldana and members of the Harvard University Research Computing team for advice in optimizing joint variant calling, Logan Blaine for the initial local run of DeNovoWEST, Vladimir Seplyarskiy and Ryan McGinty for advice regarding mutation rate models, Kaitlin Samocha for critically reviewing our manuscript, and members of the Sunyaev and Kohane research groups for helpful feedback on the manuscript text and figures. This work was funded by the National Institutes of Health (NIH) Common Fund grant U01HG007530 (to ISK), NIH National Institute of Neurological Disorders and Stroke (NINDS) grant U2CNS132415 (to ISK), NIH National Human Genome Research Institute (NHGRI) grants U01HG012009 (to SRS), R01HG012286 (to SNK & MM) and R21HG010391 (to RS), NIH National Institute of General Medical Sciences (NIGMS) grants R35GM127131 (to SRS) and 5T32GM007748 (to RR), NIH National Institute of Mental Health (NIMH) grant R01MH101244 (to SRS), NIH National Center for Advancing Translational Sciences (NCATS) grant KL2TR002552 (to RR), and NIH National Heart Lung and Blood Institute (NHLBI) grant 1R01HL164409-01 (to RS).

## Author contributions

S.N.K., M.A.M. and S.R.S. designed the study, developed and applied the statistical models, and wrote the manuscript. I.S.K. provided guidance and feedback. S.N.K. and D.T. harmonized the sequencing data, and M.B. and W.R. provided assistance. R.R. and J.K. developed the clinical evaluation protocol. R.R., J.K. and J.W. applied the clinical protocol. X.L. ran DeNovoWEST. D.B. and R.S. designed and performed the MPSA experiment, and S.N.K. analyzed the data. A.V. developed the variant browser. R.I.C., G.dV.C.N., S.F.N., and J.A.M-A. performed transcriptome sequencing and provided guidance on and ran the analysis for *MED11*. The UDN Consortia collectively evaluated patients and shared phenotype and genotype data centrally. All authors read and approved the final manuscript.

## Competing interests

The authors declare no competing interests.

## Additional information

 

[1]Department of Biomedical Informatics, Harvard Medical School, Boston, Massachusetts, USA. [2]Scripps Research Translational Institute, La Jolla, California, USA. [3]Division of Pediatric Genetics, University of California, San Diego, California, USA. [4]Department of Biostatistics, Harvard T.H. Chan School of Public Health, Boston, Massachusetts, USA. [5]Interdepartmental Neuroscience Program, Yale University, New Haven, Connecticut, USA. [6]Vrije Universiteit Amsterdam, Medical School of V, De Boelelaan 1105, 1081 HV, Amsterdam, Netherlands. [7]Division of Genetics, Department of Medicine, Brigham and Women's Hospital and Harvard Medical School, Boston, Massachusetts, USA. [8]Department of Human Genetics, David Geffen School of Medicine, University of California, Los Angeles, California, USA. [9]Department of Pathology and Laboratory Medicine, NewYork-Presbyterian Weill Cornell Medical Center, New York, New York, USA. [10]Department of Genetics, Atrius Health, Boston, Massachusetts, USA. [45]These authors contributed equally: Shilpa Nadimpalli Kobren, Mikhail A. Moldovan. ✉e-mail: shamil_sunyaev@hms.harvard.edu

## Undiagnosed Diseases Network

Jose Abdenur[11], Maria T. Acosta[12], David R. Adams[12,13], Ben Afzali[12,13], Ali Al-Beshri[14], Eric Allenspach[15], Raquel L. Alvarez[16], Justin Alvey[17], Ashley Andrews[17], Beatriz Anguiano[16], Euan A. Ashley[18], Sanaz Attaripour[11], Suha Bachir[16], Carlos A. Bacino[19], Guney Bademci[20], Ashok Balasubramanyam[19], Dustin Baldridge[21], Erin E. Baldwin[17], Allen Bale[22], Elsa Balton[15], Manisha Balwani[23], Michael Bamshad[15], Mafalda Barbosa[23], Deborah Barbouth[20], Rebekah Barrick[11], Donald Basel[24], Pinar Bayrak-Toydemir[25], Taylor Beagle[26], Alan H. Beggs[27], Edward Behrens[28], Megan Bell[26], Hugo J. Bellen[29], Paul Berger[26], Jonathan A. Bernstein[16], Gerard T. Berry[27], Louise Bier[23], Stephanie Bivona[20], Kirsten Blanco[11], Lauren Blieden[19], Elizabeth Blue[15], Devon Bonner[16], Brett Bordini[24], Nicholas Borja[20], Lorenzo Botto[17], Steven Boyden[17], Lauren C. Briere[27], Elizabeth A. Burke[12,13], Lindsay C. Burrage[19], Francisco Bustos[26], Manish J. Butte[30], Russell Butterfield[17], Peter Byers[15], William E. Byrd[31], Kaitlin Callaway[14], John Carey[17], Thomas Cassini[32], Chun-Hung Chan[26], Richard Chang[11], Sirisak Chanprasert[15], Hsiao-Tuan Chao[19], Elizabeth C. Chao[11], Ivan Chinn[19], Gary D. Clark[19], Terra R. Coakley[16], Laurel A. Cobban[27], Joy D. Cogan[32], Matthew Coggins[27], F. Sessions Cole[33], Erin Conboy[34], Brian Corner[32], Rosario I. Corona[8], William J. Craigen[19], Andrew B. Crouse[31], Vishnu Cuddapah[28], Charlotte Cunningham-Rundles[23], Precilla D'Souza[12], Hongzheng Dai[19], Nitsuh K. Dargie[15], Kahlen Darr[35], Surendra Dasari[35], Joie Davis[12,13], Margaret Delgado[12,13], Esteban C. Dell'Angelica[30], Nada Derar[22], Patricia Dickson[36], Katrina Dipple[15], Naghmeh Dorrani[30], Jessica Douglas[27], Abdul Elkadri[24], Sara Emami[16], Lisa T. Emrick[19], Christine M. Eng[37], Cecilia Esteves[38], Rachel Evard[23], Kimberly Ezell[32], Layal F. Abi Farraj[30], Elizabeth L. Fieg[27], Paul G. Fisher[16], Brent L. Fogel[30], Jiayu Fu[12,13], William A. Gahl[12,13], Rebecca Ganetzky[28], Eric Gayle[23], Bruce Gelb[23], Mark Gerstein[22], Emily Glanton[38], Ian Glass[15], Page C. Goddard[16], Joanna M. Gonzalez[20], John E. Gorzynski[16], Brett H. Graham[34], Andrea Gropman[12], Meghan C. Halley[16], Winston Halstead[22], Rizwan Hamid[32], Neil Hanchard[12,13], Kelly Hassey[28], Caroline Hendry[22], Frances High[27], Fuki M. Hisama[15], Ingrid A. Holm[27], Jason Hom[16], Martha Horike-Pyne[15], Yan Huang[12,13], Alden Huang[30], Monika Weisz Hubshman[19], Anna Hurst[14], John A. Phillips III[32], Wendy Introne[12,13], Ayuko Iverson[23], Gail P. Jarvik[15], Orpa Jean-Marie[12,13], Lauren Jeffries[22], Joanna Jen[23], Tanner D. Jensen[16], Yong-Hui Jiang[22], Vaidehi Jobanputra[39], Oguz Kanca[29], Yigit Karasozen[30], Odelya Kaufman[22], Laura Keehan[16], Shamika Ketkar[19], Dana Kiley[36], Gonench Kilich[28], Eric Klee[35], Shilpa Nadimpalli Kobren ⑩[1,45], Isaac S. Kohane[1], Jennefer N. Kohler[16], Bruce Korf[14], Susan Korrick[27], Elijah Kravets[16], Runjun Kumar[15], Seema R. Lalani[19], Brendan C. Lanpher[35], Ian R. Lanza[35], Kumarie Latchman[20], Kimberly LeBlanc[38], Brendan H. Lee[19], Miranda Leitheiser[26], Monkol Lek[22], Kathleen A. Leppig[15], Mia Levanto[16], Richard A. Lewis[19], Rachel Li[26], Khurram Liaqat[34], Pengfei Liu[37], Nicola Longo[17], Joseph Loscalzo[27], Richard L. Maas[27], Ellen F. Macnamara[12], Calum A. MacRae[27], Valerie V. Maduro[12], Rachel Mahoney[38], MayChristine V. Malicdan[12,13], Tarun K. K. Mamidi[14], Shrikant Mane[22], Lili Mantcheva[34], Rong Mao[25], Ronit Marom[19], Gabor Marth[40], Beth A. Martin[16], Martin G. Martin[30], Julian A. Martínez-Agosto[8], Shruti Marwaha[16], Taylor Maurer[16], Julie McCarrier[24], Allyn McConkie-Rosell[41],

Ashley McMinn[32], Erin McRoy[36], Hector Rodrigo Mendez[16], Matthew Might[31], Mohamad Mikati[41], Danny Miller[15], Alexander Miller[16], Ghayda Mirzaa[15], Breanna Mitchell[35], Stephen B. Montgomery[16], Paolo Moretti[17], Jennifer Morgan[26], Marie Morimoto[12,13], Tahseen Mozaffar[11], Lindsay Mulvihill[35], John J. Mulvihill[12], Michael Muriello[24], Sandesh Nagamani[19], Mariko Nakano-Okuno[31], Stanley F. Nelson[8], George de V. Carvalho Neto ®[8], Serena Neumann[32], Thomas J. Nicholas[17], Donna Novacic[12], Devin Oglesbee[35], Carol Oladele[22], James P. Orengo[19], Rebecca Overbury[17], Laura Pace[17], Stephen C. Pak[21], J. Carl Pallais[27], Neil H. Parker[30], Alex Paul[36], LéShon Peart[20], Seth Perlman[15], Leoyklang Petcharet[12,13], Jennifer E. Posey[19], Lorraine Potocki[19], Rakale C. Quarells[42], Aaron Quinlan[17], Daniel J. Rader[43], Ramakrishnan Rajagopalan[28], Deepak A. Rao[27], Anna Raper[43], Wendy Raskind[15], Adriana Rebelo[20], Chloe M. Reuter[16], Lynette Rives[32], Lance H. Rodan[27], Martin Rodriguez[14], María José Ortuño Romero[22], Jill A. Rosenfeld[19], Elisabeth Rosenthal[15], Francis Rossignol[12,13], Bianca E. Russell[30], Marla Sabaii[12,13], Mohamad Saifeddine[26], Jacinda B. Sampson[16], Suzanne Sandmeyer[11], Timothy Schedl[21], Jason Schend[26], Lisa Schimmenti[35], Kelly Schoch[41], Jennifer Schymick[16], Daryl A. Scott[19], Teodoro Jerves Serrano[22], Elaine Seto[19], Mariya Shadrina[23], Vandana Shashi[41], Emily Shelkowitz[15], Susan Shin[23], Jimann Shin[21], Saskia Shuman[23], Edwin K. Silverman[27], Giorgio Sirugo[43], Kathy Sisco[36], Tammi Skelton[14], Cara Skraban[28], Carson A. Smith[20], Kevin S. Smith[16], Jared Sninsky[19], Lilianna Solnica-Krezel[21], Ben Solomon[12,13], Albert R. La Spada[11], Michele Spencer-Manzon[22], Rebecca C. Spillmann[41], Maija-Rikka Steenari[11], Andrew Stergachis[15], Joan M. Stoler[27], Kathleen Sullivan[28], Shamil R. Sunyaev ®[1] ✉, David A. Sweetser[27], Barbara N. Pusey Swerdzewski[12], Virginia Sybert[15], Holly K. Tabor[16], Queenie Tan[35], Arjun Tarakad[19], Herman Taylor[42], Mustafa Tekin[20], Willa Thorson[20], Cynthia J. Tifft[12,13], Camilo Toro[12], Alyssa A. Tran[19], Kayla M. Treat[34], Brianna Tucker[16], Rachel A. Ungar[16], Filippo Pinto e. Vairo[35], Adeline Vanderver[28], Andres Vargas[30], Vasilis Vasiliou[22], Matt Velinder[17], James Verbsky[24], Francesco Vetrini[34], Eric Vilain[11], Dave Viskochil[17], Tiphanie P. Vogel[19], Colleen E. Wahl[12], Melissa Walker[27], Nicole M. Walley[41], Jennifer Wambach[36], Emily Wang[22], Michael F. Wangler[29], Patricia A. Ward[37], Isum Ward[26], Alistair Ward[17], Stephanie M. Ware[34], Daniel Wegner[36], Corrine K. Welt[17], Mark Wener[15], Monte Westerfield[44], Matthew T. Wheeler[16], Jordan Whitlock[31], Laurens Wiel[16], Brandon M. Wilk[14], Lynne A. Wolfe[12,13], Heidi Wood[12,13], Kim Worley[19], Elizabeth A. Worthey[14], Changrui Xiao[11], Hua Xu[22], Shinya Yamamoto[29], Hui Zhang[22], Michael Zimmermann[24] & Stephan Zuchner[20]

[11]University of California Irvine (UCI)/Children's Hospital of Orange County (CHOC) clinical site, Irvine, California, USA. [12]National Institutes of Health (NIH) Undiagnosed Diseases Program (UDP) clinical site, Bethesda, Maryland, USA. [13]National Institutes of Health (NIH) National Human Genome Research Institute (NHGRI), Bethesda, Maryland, USA. [14]University of Alabama at Birmingham clinical site, Birmingham, Alabama, USA. [15]Pacific Northwest clinical site, Seattle, Washington, USA. [16]Stanford University clinical site, Palo Alto, California, USA. [17]University of Utah clinical site, Salt Lake City, Utah, USA. [18]Stanford University Data Management and Coordinating Center, Palo Alto, California, USA. [19]Baylor College of Medicine clinical site, Houston, Texas, USA. [20]University of Miami clinical site, Miami, Florida, USA. [21]Washington University at Saint Louis, Model Organisms Screening Center, Saint Louis, Missouri, USA. [22]Yale clinical site, New Haven, Connecticut, USA. [23]Mount Sinai clinical site, New York, New York, USA. [24]Medical College of Wisconsin, Central Washington Campus clinical site, Milwaukee, Wisconsin, USA. [25]Associated Regional and University Pathologists, Inc. (ARUP Laboratories) at the University of Utah clinical site, Salt Lake City, Utah, USA. [26]Sanford Health clinical site, Sioux Falls, South Dakota, USA. [27]Harvard University clinical site, Boston, Massachusetts, USA. [28]Children's Hospital of Philadelphia clinical site, Philadelphia, Pennsylvania, USA. [29]Baylor College of Medicine Model Organisms Screening Center, Houston, Texas, USA. [30]University of California at Los Angeles clinical site, Los Angeles, California, USA. [31]University of Alabama at Birmingham, Data Management and Coordinating Center, Birmingham, Alabama, USA. [32]Vanderbilt University clinical site, Nashville, Tennessee, USA. [33]Washington University at Saint Louis, Data Management and Coordinating Center, St. Louis, Missouri, USA. [34]Iowa University clinical site, Iowa City, Iowa, USA. [35]Mayo Clinic, Rochester, Minnesota, USA. [36]Washington University at Saint Louis, clinical site, St. Louis, Missouri, USA. [37]Baylor College of Medicine Sequencing Core, Houston, Texas, USA. [38]Harvard University Data Management and Coordinating Center, Boston, Massachusetts, USA. [39]Columbia University clinical site, New York, New York, USA. [40]University of Utah Data Management and Coordinating Center, Salt Lake City, Utah, USA. [41]Duke University clinical site, Durham, North Carolina, USA. [42]Morehouse Data Management and Coordinating Center, Atlanta, Georgia, USA. [43]Children's Hospital of Philadelphia and University of Pennsylvania clinical site, Philadelphia, Pennsylvania, USA. [44]University of Oregon Model Organisms Screening Center, Eugene, Oregon, USA.

