## [Transparent Peer Review file · Nature Communications]

Joint, multifaceted genomic analysis enables diagnosis of diverse, ultra-rare monogenic presentations

Corresponding Author: Professor Shamil Sunyaev

Version 0:

Reviewer comments:

Reviewer #4

(Remarks to the Author)

This is a well-written study with some interesting insights for those of us in the RD diagnostic space. I am giving feedback on the response to reviewer 1.

[R1.1] It was unclear to me why the authors did not compare the results of both methods on the original DDD+GeneDx+RUMC cohort? The authors show the results of the comparison as a supplementary figure but it would be better to also have a supplementary table with the exact results of the analyses and comparison.

- Our original Supplementary Figure S16 depicts the relative abilities of DeNovoWEST and RaMeDiES-DN (our de novo recurrence method, using only CADD deleteriousness prediction scores for fair comparison) to recover known autosomal dominant genes (as listed in OMIM) on de novos from the original DDD+GeneDx+RUMC cohort. Note that we subsetting the starting list of de novos to exclude chromosome X and variants that could not be accurately left-aligned and lifted over to GRCh38. We have now added Supplementary Table S5 with the RaMeDiES-DN gene rankings (by Q-value) and DeNovoWEST gene rankings (by P-value) with additional columns indicating autosomal dominant status in OMIM as well as how the gene was identified in the original DeNovoWEST Nature 2020 publication.

- The exact results of the analyses for this comparison are now a lot clearer with this new supplementary table.

[R1.2] Similarly, the authors indicate that the very strength of their method is that it allows to easily combine datasets at the level of mutation counts. This is indeed a major advantage, but it would be much more convincing if the authors actually did this in their study. Several large studies are available with identified de novo mutations that would allow the authors to demonstrate the strength of their method, with the potential to identify many more diseases at high confidence levels (including the GeneDX+DDD+RUMC dataset that is already used by the authors).

- As a proof of concept, we divided the GeneDx+DDD+RUMC dataset into three cohorts by data source. We ran RaMeDiES-DN to produce shareable summary-level mutational target statistics on each of these three subsets in turn, and then combined the intermediate results to produce a final ranked list (Supplementary Table S6). We compared this meta-analysis run to the results obtained from running RaMeDiES-DN on the full cohort of DDD+GeneDx+RUMC variant- and patient-level data as well (Supplementary Table S5). We uncovered the same genes with significance across both versions. We provide the summary-level mutational target files for DDD, GeneDx, RUMC, and UDN in our GitHub repository.

- This analysis nicely demonstrates that combining summary-level statistics works just as well as integrating the data from the very beginning and will make the software more applicable in multiple, real life future scenarios.

[R1.3] The authors identified LRRC7 as a novel candidate NDD gene in their cohort. I note that very recently this gene was also identified in another study: <https://www.nature.com/articles/s41467-024-52095-x>

- Yes, we had been in touch with the UDN clinical team(s) about this finding over a year ago, and it is unsurprising that their

parallel work collaborating on additional hits from Matchmaker Exchange has now been published. We have updated the text accordingly and included this new citation in our manuscript.

- It is good to see this independent confirmation and the authors now cite this work.

[R1.4] I agree that homozygous variants are rare in outbred populations, but in practice one does not obtain only samples from a single outbred population and cryptic consanguinity does occur, and the method would miss relatively straightforward variants. I do not understand the methodological issues concerning this, but I can agree with the authors that overall this may be a relatively minor concern.

- We have added functionality to our codebase to enable analysis of all biallelic variants (including homozygous recessive variants) and updated our text accordingly. The issue with including homozygous recessive variants from consanguineous families is theoretical, not technical. Specifically, the assumption that rare variants inherited from each parent stem from independent mutational events is violated in the case of consanguinity, where the same variant inherited from each parent likely arose from the same single ancestral mutational event. This explanation was originally in a supplementary note, but has now been reiterated in the main text and in our GitHub repository. We have also added a clarifying sentence to the Limitations section of our Discussion that our rigorous statistical framework prioritizes precision over comprehensiveness by design.

- The addition of recessive, homozygous variant analysis to the software is a welcome addition, along with the explanation of the limitations of this as the inherited variants could well not be independent events.

[R1.6] My initial comment about not leveraging the whole genome nature of the cohort referred to variants outside of genic regions, as well as CNVs/SVs/STRs.

- This is a fair concern, and one that is described in our limitations section. The genetics community has not yet developed as accurate a mutational model nor pathogenicity predictions for CNVs/SVs/STRs as we have for SNV/indels (note that we developed a separate mutational model to include short indels). Similarly, our approach relies on quality pathogenicity predictions, which are still lacking for intergenic variants.

- I agree that for intergenic variants, the available prediction algorithms have a much more limited accuracy and leaving this out and discussing it as limitation is appropriate.

[R1.7] I think it is good practice to publish all de novo mutations of the cohort as a supplement to the paper for reproducibility and future use by others.

- Agreed, thank you for this suggestion. We have now published the list of de novo mutations found across the cohort, using transformed, deidentified patient IDs that are also used in our public-facing browser (Supplementary Table S2). Our summary-level mutational targets for the UDN cohort, computed from the set of starting de novos, are also now available in our GitHub repository.

- This is a good addition that many in the community will find useful, especially if the variants are eventually classified and make their way into Clinvar etc.

(Remarks on code availability)
